# Computationally and statistically efficient learning of causal Bayes nets using path queries

**Kevin Bello**
Department of Computer Science
Purdue University
West Lafayette, IN, USA
kbellome@purdue.edu

**Jean Honorio**
Department of Computer Science
Purdue University
West Lafayette, IN, USA
jhonorio@purdue.edu

## Abstract

Causal discovery from empirical data is a fundamental problem in many scientific domains. Observational data allows for identifiability only up to Markov equivalence class. In this paper we first propose a polynomial time algorithm for learning the exact correctly-oriented structure of the transitive reduction of any causal Bayesian network with high probability, by using *interventional path queries*. Each path query takes as input an origin node and a target node, and answers whether there is a directed path from the origin to the target. This is done by *intervening* on the origin node and observing samples from the target node. We theoretically show the logarithmic sample complexity for the size of interventional data per path query, for continuous and discrete networks. We then show how to learn the *transitive* edges using also logarithmic sample complexity (albeit in time exponential in the maximum number of parents for discrete networks), which allows us to learn the full network. We further extend our work by reducing the number of interventional path queries for learning rooted trees. We also provide an analysis of imperfect interventions.

## 1 Introduction

**Motivation.** Scientists in diverse areas (e.g., epidemiology, economics, etc.) aim to unveil causal relationships within variables from collected data. For instance, biologists try to discover the causal relationships between genes. By providing a specific treatment to a particular gene (origin), one can observe whether there is an effect in another gene (target). This effect can be either direct (if the two genes are connected with a directed edge) or indirect (if there is a directed path from the origin to the target gene).

Bayesian networks (BNs) are powerful representations of joint probability distributions. BNs are also used to describe causal relationships among variables [14]. The structure of a *causal* BN (CBN) is represented by a directed acyclic graph (DAG), where nodes represent random variables, and an edge between two nodes $X$ and $Y$ (i.e., $X \to Y$) represents that the former ($X$) is a direct cause of the latter ($Y$). Learning the DAG structure of a CBN is of much relevance in several domains, and is a problem that has long been studied during the last decades.

From *observational* data alone (i.e., *passively* observed data from an undisturbed system), DAGs are only identifiable up to Markov equivalence.[1] However, since our goal is causal discovery, this is inadequate as two BNs might be Markov equivalent and yet make different predictions about the

consequences of interventions (e.g., $X \leftarrow Y$ and $X \rightarrow Y$ are Markov equivalent, but make very different assertions about the effect of changing $X$ on $Y$). In general, the only way to distinguish causal graphs from the same Markov equivalence class is to use *interventional* data [10, 11, 19]. This data is produced after performing an experiment (intervention) [21], in which one or several random variables are forced to take some specific values, irrespective of their causal parents.

**Related work.** Several methods have been proposed for learning the structure of Bayesian networks from *observational* data. Approaches ranging from score-maximizing heuristics, exact exponential-time score-maximizing, ordering-based search methods using MCMC, and test-based methods have been developed to name a few. The umbrella of tools for structure learning of Bayesian networks go from exact methods (exponential-time with convergence/consistency guarantees) to heuristics methods (polynomial-time without any convergence/consistency guarantee). [12] provide a score-maximizing algorithm that is likelihood consistent, but that needs super-exponential time. [27, 3] provide polynomial-time test-based methods that are structure consistent, but results hold only in the infinite-sample limit (i.e., when given an infinite number of samples). [5] show that greedy hill-climbing is structure consistent in the infinite sample limit, with unbounded time. [34] show structure consistency of a single network and do not provide uniform consistency for all candidate networks (the authors discuss the issue of not using the union bound in their manuscript). From the *active learning* literature, most of the works first find a Markov equivalence class (or assume that they have one) from purely *observational* data and then orient the edges by using as few *interventions* as possible. [19, 28] propose an exponential-time Bayesian approach relying on structural priors and MCMC. [10, 11, 25] present methods to find an optimal set of interventions in polynomial time for a class of chordal DAGs. Unfortunately, finding the initial Markov equivalence class remains exponential-time for general DAGs [4, 21]. [7] propose an exponential-time dynamic programming algorithm for learning DAG structures exactly. [29] propose a constraint-based method to combine heterogeneous (observational and interventional) datasets but rely on solving instances of the (NP-hard) *boolean satisfiability problem*. [8] analyzed the number of interventions sufficient and in the worst-case necessary to determine the structure of any DAG, although no algorithm or sample complexity analysis was provided. Literature on learning *structural equation models* from observational data, include the work on continuous [23, 26] and discrete [22] additive noise models. Correctness was shown for the continuous case [23] but only in the infinite-sample limit. [13] propose a method to learn the exact observable graph by using $\mathcal{O}(\log n)$ multiple-vertex interventions, where $n$ is the number of variables, through the use of pairwise conditional independence test and assuming access to the post-interventional graph. However the size of the intervened set is $\mathcal{O}(n/2)$ which leads to a $\mathcal{O}(2^{n/2})$ number of experiments in the worst case. In contrast to this work, we perform single-vertex interventions as a first step and then multiple-vertex interventions while keeping a small sample complexity. While this increments the number of interventions to $n$, we have a better control of the number of experiments.

*Remark* 1. In this paper we consider one intervention as one selection of variables to intervene. However, we consider an experiment as the actual setting of values to the variables. For example, if a variable $X$ takes $p$ different values, then one experiment is $X$ taking one specific value. To intervene one binary variable, it is common to make 2 experiments, one under treatment, and one under no treatment.

For a discussion of learning from purely interventional data, as well as availability of purely interventional data, see Appendix A.

**Contributions.** We propose a polynomial time algorithm with provable guarantees for exact learning of the transitive reduction of any CBN by using interventional path queries. We emphasize that modeling the problem of structure learning of CBNs as a problem of reconstructing a graph using path queries is also part of our contributions. We analyze the sample complexity for answering every interventional path query and show that for CBNs of discrete random variables with maximum domain size $r$, the sample complexity is $\mathcal{O}(\log(nr))$; whereas for CBNs of sub-Gaussian random variables, the sample complexity is $\mathcal{O}(\sigma_{ub}^2 \log n)$ where $\sigma_{ub}^2$ is an upper bound of the variable variances (marginally as well as after interventions). Then, we introduce a new type of query to learn the *transitive edges* (i.e., the edges that are part of the true network but not of the transitive reduction), while the learning is not in polynomial-time for discrete CBNs in the worst case (exponential in the maximum number of parents), we show that the sample complexity is still polynomial. We also present two extensions: for learning rooted trees the number of path queries is reduced to $\mathcal{O}(n \log n)$,

which is an improvement from the $n^2$ for general DAGs. We also provide an analysis of imperfect interventions. We summarize our main results in Table 1 and compare them to one of the closest related work [13].

Table 1: Here $n$ is the number of variables, $\sigma_{ub}^2$ is an upper bound of the variable variances (marginally as well as after interventions), $t$ is the maximum number of parents, $r$ is the maximum number of values a discrete variable can take, and $B$ denotes the time complexity of an independence-test oracle. Note that $B \in \mathcal{O}(2^n)$ in the worst case and not $\mathcal{O}(2^t)$ because [13] can select an intervention set of $n/2$ nodes (see for example Appendix F.2). In this table, *novel* indicates that no prior work provided results on the respective subject. Finally, C and D denote continuous and discrete variables respectively.

| Graph | Var. | Algorithms | Sample complexity | Time complexity |
|-------|------|------------|-------------------|-----------------|
| General DAGs | D | 1, 5, 3, 7 (our work) | $\mathcal{O}(n^2 2^t \log(nr))$ (Novel, see Thms. 1, 3) | $\mathcal{O}(n^2 2^t \log(nr))$ |
| | | 1, 3 in [13] | - | $\mathcal{O}(Btn^2 \log^2 n)$ ($B \in \mathcal{O}(2^n)$) |
| | C | 1, 6, 3, 8 (our work) | $\mathcal{O}(n^2 \sigma_{ub}^2 \log n)$ (Novel, see Thms. 2, 4) | $\mathcal{O}(n^2 \sigma_{ub}^2 \log n)$ |
| Rooted trees | D | See Section 4 | $\mathcal{O}(n \log^2(nr))$ (Novel, see Section 4) | $\mathcal{O}(n \log^2(nr))$ |

| Graph | Var. | Algorithms | # of interventions | # of experiments |
|-------|------|------------|--------------------|------------------|
| General DAGs | D | 1, 5, 3, 7 (our work) | $\mathcal{O}(n^2)$ | $\mathcal{O}(n^2 2^t)$ |
| | | 1, 3 in [13] | $\mathcal{O}(\log n)$ | $\mathcal{O}(2^n \log n)$ (see Appendix F.2.) |
| | C | 1, 6, 3, 8 (our work) | $\mathcal{O}(n^2)$ | $\mathcal{O}(n^2)$ |
| Rooted trees | D | See Section 4 | $\mathcal{O}(n)$ | $\mathcal{O}(nr)$ |

## 2 Preliminaries

In this section, we introduce our formal definitions and notations. Vectors and matrices are denoted by lowercase and uppercase bold faced letters respectively. Random variables are denoted by italicized uppercase letters and their values by lowercase italicized letters. Vector $\ell_p$-norms are denoted by $\|\cdot\|_p$. For matrices, $\|\cdot\|_{p,q}$ denotes the entrywise $\ell_{p,q}$ norm, i.e., for $\|\mathbf{A}\|_{p,q} = \|(\|(A_{1,1}, \ldots, A_{m,1})\|_p, \ldots, \|(A_{1,n}, \ldots, A_{m,n})\|_p)\|_q$.

Let $G = (V, E)$ be *directed acyclic graph* (DAG) with vertex set $V = \{1, \ldots, n\}$ and edge set $E \subset V \times V$, where $(i, j) \in E$ implies the edge $i \to j$. For a node $i \in V$, we denote $\pi_G(i)$ as the parent set of the node $i$. In addition, a directed path of length $k$ from node $i$ to node $j$ is a sequence of nodes $(i, v_1, v_2, \ldots, v_{k-1}, j)$ such that $\{(i, v_1), (v_1, v_2), \ldots, (v_{k-2}, v_{k-1}), (v_{k-1}, j)\}$ is a subset of the edge set $E$.

Let $\boldsymbol{X} = \{X_1, \ldots, X_n\}$ be a set of random variables, with each variable $X_i$ taking values in some domain $Dom[X_i]$. A *Bayesian network* (BN) over $\boldsymbol{X}$ is a pair $\mathcal{B} = (G, \mathcal{P}_G)$ that represents a distribution over the joint space of $\boldsymbol{X}$. Here, $G$ is a DAG, whose nodes correspond to the random variables in $\boldsymbol{X}$ and whose structure encodes conditional independence properties about the joint distribution, while $\mathcal{P}_G$ quantifies the network by specifying the *conditional probability distributions* (CPDs) $P(X_i|\boldsymbol{X}_{\pi_G(i)})$. We use $\boldsymbol{X}_{\pi_G(i)}$ to denote the set of random variables which are parents of $X_i$. A Bayesian network represents a *joint probability distribution* over the set of variables $\boldsymbol{X}$, i.e., $P(X_1, \ldots, X_n) = \prod_{i=1}^n P(X_i|\boldsymbol{X}_{\pi_G(i)})$.

Viewed as a probabilistic model, a BN can answer any "conditioning" query of the form $P(\boldsymbol{Z}|\boldsymbol{E} = \boldsymbol{e})$ where $\boldsymbol{Z}$ and $\boldsymbol{E}$ are sets of random variables and $\boldsymbol{e}$ is an assignment of values to $\boldsymbol{E}$. Nonetheless, a BN can also be viewed as a *causal model* or causal BN (CBN) [21]. Under this perspective, the CBN can also be used to answer *interventional* queries, which specify probabilities after we intervene in the model, forcibly setting one or more variables to take on particular values. The manipulation theorem [27, 21] states that one can compute the consequences of such interventions (perfect interventions) by "cutting" all the arcs coming into the nodes which have been clamped by intervention, and then doing typical probabilistic inference in the "mutilated" graph (see Figure 1 as an example). We follow the standard notation [21] for denoting the probability distribution of a variable $X_j$ after intervening $X_i$, that is, $P(X_j|do(X_i = x_i))$. In this case, the joint distribution after intervention is given by $P(X_1, \ldots, X_{i-1}, X_{i+1}, \ldots, X_n|do(X_i = x_i)) = \mathbb{1}[X_i = x_i] \prod_{j \neq i} P(X_j|\boldsymbol{X}_{\pi_G(j)})$.

We refer to CBNs in which all random variables $X_i$ have finite domain, $Dom[X_i]$, as discrete CBNs. In this case, we will denote the probability mass function (PMF) of a random variable as a vector.

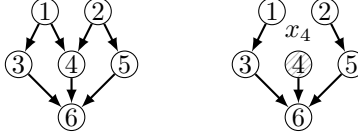

Figure 1: (Left) A CBN of 6 variables, where the joint distribution, $P(\boldsymbol{X})$, is factorized as $\prod_i P(X_i|\boldsymbol{X}_{\pi_{\mathrm{G}}(i)})$. (Right) The mutilated CBN after intervening $X_4$ with value $x_4$. Note that the edges $\{(1,4),(2,4)\}$ are not part of the CBN after the intervention, thus, the new joint is $P(\boldsymbol{X}|do(X_4 = x_4)) = \mathbb{1}[X_4 = x_4]\prod_{i\neq 4} P(X_i|\boldsymbol{X}_{\pi_{\mathrm{G}}(i)})$.

That is, a PMF, $P(Y)$, can be described as a vector $\mathbf{p}(Y) \in [0,1]^{|Dom[Y]|}$ indexed by the elements of $Dom[Y]$, i.e., $\mathrm{p}_j(Y) = P(Y=j), \forall j \in Dom[Y]$. We refer to networks with variables that have continuous domains as continuous CBNs.

Next, we formally define transitive edges.

**Definition 1** (Transitive edge). *Let* $\mathrm{G} = (\mathrm{V},\mathrm{E})$ *be a DAG. We say that an edge* $(i,j) \in \mathrm{E}$ *is transitive if there exists a directed path from* $i$ *to* $j$ *of length greater than 1.*

The algorithm for removing transitive edges from a DAG is called *transitive reduction* and it was introduced in [1]. The transitive reduction of a DAG G, $\mathrm{TR}(\mathrm{G})$, is then G without any of its transitive edges. Our proposed methods also make use of *path queries*, which we define as follows:

**Definition 2** (Path query). *Let* $\mathrm{G} = (\mathrm{V},\mathrm{E})$ *be a DAG. A path query is a function* $Q_{\mathrm{G}} : \mathrm{V} \times \mathrm{V} \to \{0,1\}$ *such that* $Q_{\mathrm{G}}(i,j) = 1$ *if there exists a directed path in* G *from* $i$ *to* $j$*, and* $Q_{\mathrm{G}}(i,j) = 0$ *otherwise.*

**General DAGs are identifiable only up to their transitive reduction by using path queries.** In general, DAGs can be non-identifiable by using path queries. We will use $Q(i,j)$ to denote $Q_{\mathrm{G}}(i,j)$ since for our problem, the DAG G is fixed (but unknown). For instance, consider the two graphs shown in Figure 2. In both cases, we have that $Q(1,2) = Q(1,3) = Q(2,3) = 1$. Thus, by using path queries, it is impossible to discern whether the edge $(1,3)$ exists or not. Later in Subsection 3.3 we focus on the recovery of transitive edges, which requires a different type of query.

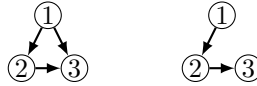

Figure 2: Two directed acyclic graphs that produce the same answers when using path queries.

How to answer path queries is a key step in this work. Since we answer path queries by using a finite number of interventional samples, we require a noisy path query, which is defined below.

**Definition 3** ($\delta$-noisy partially-correct path query). *Let* $\mathrm{G} = (\mathrm{V},\mathrm{E})$ *be a DAG, and let* $Q_{\mathrm{G}}$ *be a path query. Let* $\delta \in (0,1)$ *be a probability of error. A $\delta$-noisy partially-correct path query is a function* $\tilde{Q}_{\mathrm{G}} : \mathrm{V} \times \mathrm{V} \to \{0,1\}$ *such that* $\tilde{Q}_{\mathrm{G}}(i,j) = Q_{\mathrm{G}}(i,j)$ *with probability at least* $1 - \delta$ *if* $i \in \pi_{\mathrm{G}}(j)$ *or if there is no directed path from* $i$ *to* $j$*.*

We will use the term noisy path query to refer to $\delta$-noisy partially-correct path query. Note that Definition 3 requires a noisy path query to be correct *only in certain cases*, when one variable is parent of the other, or when there is no directed path between them. We do not require correctness when there is a directed path between $i$ and $j$ and $i$ is not a parent of $j$, that is, when the path length is greater than 1. Note that the uncertainty of the exact recovery of the transitive reduction relies on answering multiple noisy path queries.

## 2.1 Assumptions

Here we state the main set of assumptions used throughout our paper.

**Assumption 1.** *Let* $\mathrm{G} = (\mathrm{V},\mathrm{E})$ *be a DAG. All nodes in* G *are observable, furthermore, we can perform interventions on any node* $i \in \mathrm{V}$*.*

**Assumption 2** (Causal Markov). *The data is generated from an underlying CBN* $(G, \mathcal{P}_G)$ *over* $\boldsymbol{X}$.

**Assumption 3** (Faithfulness). *The distribution $P$ over $\boldsymbol{X}$ induced by $(G, \mathcal{P}_G)$ satisfies no independences beyond those implied by the structure of $G$. We also assume faithfulness in the post-interventional distribution.*

Assumption 1 implies the availability of purely interventional data, and has been widely used in the literature [19, 28, 11, 10, 25, 13]. We consider only observed variables because we perform interventions on each node, thus, our method is robust to latent confounders. (See Appendix E for more details). With Assumption 2, we assume that any population produced by a causal graph has the independence relations obtained by applying d-separation to it, while with Assumption 3, we ensure that the population has exactly these and no additional independences [27, 28, 25, 11, 29].

## 3 Algorithms and Sample Complexity

Next, we present our first set of results and provide a formal analysis on the sample complexity.

### 3.1 Algorithm for Learning the Transitive Reduction of CBNs

[13] show that by using $\mathcal{O}(\log n)$ multiple-vertex interventions, one can recover the transitive reduction of a DAG. However, in this case, each set of intervened variables has a size of $\mathcal{O}(n/2)$, which means that the method of [13] has to perform a total of $\mathcal{O}(2^{n/2} \log n)$ experiments, one for each possible setting of the $\mathcal{O}(n/2)$ intervened variables (see an example of this in Appendix D). Thus, in this part we work with single-vertex interventions to avoid the exponential number of experiments. We can then learn the transitive reduction as follows (see mote details in Appendix B.1).

**Algorithm 1.** *Start with a set of edges $\hat{E} = \varnothing$. Then for each pair of nodes $i, j \in V$, compute the noisy path query $\tilde{Q}(i, j)$ and add the edge $(i, j)$ to $\hat{E}$ if the query returns $1$. Finally, compute the transitive reduction of $\hat{E}$ in poly-time [1], and return $\hat{E}$.*

As seen in the next section, each query is computed using single-vertex interventions. In fact, for each intervened node, we can compute $n$ queries, i.e., while the number of queries is $n^2$, the number of interventions is $n$. This number of single-vertex interventions is necessary in the worst case [8].

It is natural to ask what would be the benefit of using path queries. A query $\tilde{Q}(i, j)$ can be interpreted as observing the variable $X_j$ after intervening $X_i$. Under this viewpoint, if one could reduce the number of queries for learning certain classes of graphs, then not only might the number of interventions decrease but the number of variables to observe too. That is, if one knows a priori that the topology of the graph belongs to a certain family of graphs then it may be possible to reduce the number of queries (see for example Section 4). This is important in practice as both performing interventions and observing variables might be costly. We first focus in learning general DAGs, in which a number of $\Omega(n^2)$ path queries is in the worst case necessary for any conceivable algorithm. (See Theorems 7 and 8 in [32]). Later we show that a number of $\mathcal{O}(n \log n)$ noisy path queries[2] suffices for learning rooted trees.

### 3.2 Noisy Path Query Algorithm

The next two propositions are important for answering a path query.

**Proposition 1.** *Let $\mathcal{B} = (G, \mathcal{P}_G)$ be a CBN with $X_i, X_j \in \boldsymbol{X}$ being any two random variables in $G$. If there is no directed path from $i$ to $j$ in $G$, then $P(X_j | do(X_i = x_i)) = P(X_j)$.*

**Proposition 2.** *Let $\mathcal{B} = (G, \mathcal{P}_G)$ be a CBN and let $X_i$ and $X_j$ be two random variables in $G$, such that $i \in \pi_G(j)$. Then, there exists $x_i$ and $x_i'$ such that:*

$$1.\ P(X_j) \neq P(X_j | do(X_i = x_i))\ \text{ and }\ 2.\ P(X_j | do(X_i = x_i)) \neq P(X_j | do(X_i = x_i'))$$

See Appendix F for details of all proofs. Proposition 2 motivates the idea that we can search for two different values of $X_i$ to determine the causal dependence on $X_j$ (Claim 2), which is

arguably useful for discrete CBNs. Alternatively, we can use the expected value of $X_j$, since $\mathbb{E}[X_j] \neq \mathbb{E}[X_j|do(X_i = x_i)]$ implies that $P(X_j) \neq P(X_j|do(X_i = x_i))$ (Claim 1).

Next, we propose a polynomial time algorithm for answering a noisy path query. Algorithm 2 presents the procedure in an intuitive way. Here, the type of statistic is motivated by Lemmas 1 and 2, and the value of interventions and threshold $t$ are motivated by Theorems 1 and 2. See Appendix B.2 (Algorithms 5 and 6) for the specific details of the algorithms for discrete and continuous CBNs.

---

**Algorithm 2** Noisy path query algorithm

---

**Input:** Nodes $i$ and $j$, number of interventional samples $m$, and threshold $t$.
**Output:** $\tilde{Q}(i,j)$
  1: Intervene $X_i$ by setting its value to $x_i \in Dom[X_i]$, and observe $m$ samples of $X_j$
  2: Compute a statistic of $X_j$ and return 1 if it is greater than $t$.

---

**Discrete random variables.** In this paper we use conditional probability tables (CPTs) as the representation of the CPDs for discrete CBNs. Next, we present a theorem that provides the sample complexity of a noisy path query.

**Theorem 1.** *Let $\mathcal{B} = (\mathrm{G}, \mathcal{P}_\mathrm{G})$ be a discrete CBN, such that each random variable $X_j$ has a finite domain $Dom[X_j]$, with $\big|Dom[X_j]\big| \leq r$. Furthermore, let*

$$\gamma = \min_{\substack{j \in \mathrm{V} \\ i \in \pi_\mathrm{G}(j)}} \min_{\substack{x_i, x_i' \in Dom[X_i] \\ \mathbf{p}(X_j|do(X_i=x_i)) \neq \mathbf{p}(X_j|do(X_i=x_i'))}} \|\mathbf{p}(X_j|do(X_i = x_i)) - \mathbf{p}(X_j|do(X_i = x_i'))\|_\infty,$$

*and let $\hat{\mathrm{G}} = (\mathrm{V}, \hat{\mathrm{E}})$ be the learned graph by using Algorithm 1. Then for $\gamma > 0$ and a fixed probability of error $\delta \in (0,1)$, we have $P\left(\mathtt{TR}(\mathrm{G}) = \hat{\mathrm{G}}\right) \geq 1 - \delta$, provided that $m \in \mathcal{O}(\frac{1}{\gamma^2}\left(\ln n + \ln \frac{r}{\delta}\right))$ interventional samples are used per $\delta$-noisy partially-correct path query in Algorithm 5.*

Intuitively, the value $\gamma$ characterizes the *minimum causal effect* among all the pair of parent-child nodes. Due to Assumption 3, and the fact that an edge represents a causal relationship, we have $\gamma > 0$. This value is used for deciding whether two empirical PMFs are equal or not in our path query algorithm (Algorithm 5), which implements Claim 2 in Proposition 2. Finally, in practice, the value of $\gamma$ is unknown[3]. Fortunately, knowing a lower bound of $\gamma$ suffices for structure recovery.

**Continuous random variables.** For continuous CBNs, our algorithm compares two empirical expected values for answering a path query. This is related to Claim 1 in Proposition 2, since $\mathbb{E}[X_j] \neq \mathbb{E}[X_j|do(X_i = x_i)]$ implies $P(X_j) \neq P(X_j|do(X_i = x_i))$. We analyze continuous CBNs where every random variable is sub-Gaussian. The class of sub-Gaussian variates includes for instance Gaussian variables, any bounded random variable (e.g., uniform), any random variable with strictly log-concave density, and any finite mixture of sub-Gaussian variables. Note that sample complexity using sub-Gaussian variables has been studied in the past for other models, such as Markov random fields [24]. Next, we present a theorem that formally characterizes the class of continuous CBNs that our algorithm can learn, and provides the sample complexity for each noisy path query.

**Theorem 2.** *Let $\mathcal{B} = (\mathrm{G}, \mathcal{P}_\mathrm{G})$ be a continuous CBN such that each variable $X_j$ is a sub-Gaussian random variable with full support on $\mathbb{R}$, with mean $\mu_j = 0$ and variance $\sigma_j^2$. Let $\mu_{j|do(X_i=z)}$ and $\sigma_{j|do(X_i=z)}^2$ denote the expected value and variance of $X_j$ after intervening $X_i$ with value $z$, assuming also that the variables remain sub-Gaussian after performing an intervention. Furthermore, let*

$$\mu(\mathcal{B}, z) = \min_{j \in \mathrm{V}, i \in \pi_\mathrm{G}(j)} \left|\mu_{j|do(X_i=z)}\right|, \quad \sigma^2(\mathcal{B}, z) = \max\left(\max_{j \in \mathrm{V}, i \in \pi_\mathrm{G}(j)} \sigma_{j|do(X_i=z)}^2, \max_{j \in \mathrm{V}} \sigma_j^2\right),$$

*and let $\hat{\mathrm{G}} = (\mathrm{V}, \hat{\mathrm{E}})$ be the learned graph by using Algorithm 1. If there exist an upper bound $\sigma_{ub}^2$ and a finite value $z$ such that $\sigma^2(\mathcal{B}, z) \leq \sigma_{ub}^2$ and $\mu(\mathcal{B}, z) \geq 1$, then for a fixed probability of error $\delta \in (0,1)$, we have $P\left(\mathtt{TR}(\mathrm{G}) = \hat{\mathrm{G}}\right) \geq 1 - \delta$, provided that $m \in \mathcal{O}(\sigma_{ub}^2 \log \frac{n}{\delta})$ interventional samples are used per $\delta$-noisy partially-correct path query in Algorithm 6.*

Note that the conditions $\mu_j = 0, \forall j \in V$, and $\mu(\mathcal{B}, z) \geq 1$ are set to offer clarity in the derivations. One could for instance set an upper bound for the magnitude of $\mu_j$, assume $\mu(\mathcal{B}, z)$ to be greater than this upper bound plus 1, and still have the same sample complexity. Finally, our motivation for giving such conditions is that of guaranteeing a proper separation of the expected values in cases where there is effect of a variable $X_i$ over another variable $X_j$, versus cases where there is no effect at all.

Next, we define the additive sub-Gaussian noise model (ASGN).

**Definition 4.** *Let* $G = (V, E)$ *be a DAG, let* $\mathbf{W} \in \mathbb{R}^{n \times n}$ *be the matrix of edge weights and let* $\mathcal{S} = \{\sigma_i^2 \in \mathbb{R}_+ | i \in V\}$ *be the set of noise variances. An additive sub-Gaussian noise network is a tuple* $(G, \mathcal{P}(\mathbf{W}, \mathcal{S}))$ *where each variable* $X_i$ *can be written as follows:* $X_i = \sum_{j \in \pi_G(i)} W_{ij} X_j + N_i, \forall i \in V$, *with* $N_i$ *being an independent sub-Gaussian noise with full support on* $\mathbb{R}$, *with zero mean and variance* $\sigma_i^2$ *for all* $i \in V$, *and* $W_{ij} \neq 0$ *iff* $(j, i) \in E$.

*Remark* 2. Let $\mathcal{B} = (G, \mathcal{P}(\mathbf{W}, \mathcal{S}))$ be an ASGN network. We can rewrite the model in vector form as: $\mathbf{x} = \mathbf{W}\mathbf{x} + \mathbf{n}$ or equivalently $\mathbf{x} = (\mathbf{I} - \mathbf{W})^{-1}\mathbf{n}$, where $\mathbf{x} = (X_1, \ldots, X_n)$ and $\mathbf{n} = (N_1, \ldots, N_n)$ are the vector of random variables and the noise vector respectively. Additionally, we denote $\odot_i \mathbf{W}$ as the weight matrix $\mathbf{W}$ with its $i$-th row set to 0. This means that we can interpret $\odot_i \mathbf{W}$ as the weight matrix after performing and intervention on node $i$ (mutilated graph).

We now present a corollary that fulfills the conditions presented in Theorem 2.

**Corollary 1** (Additive sub-Gaussian noise model). *Let* $\mathcal{B} = (G, \mathcal{P}(\mathbf{W}, \mathcal{S}))$ *be an ASGN network as in Definition 4, such that* $\sigma_j^2 \leq \sigma_{max}^2, \forall j \in V$. *Also, let* $w_{min} = \min_{(i,j) \in E} |\{(\mathbf{I} - \odot_i \mathbf{W})^{-1}\}_{ji}|$, *and* $w_{max} = \max(\|(\mathbf{I} - \mathbf{W})^{-1}\|_{\infty,2}^2, \max_{i \in V}\|(\mathbf{I} - \odot_i \mathbf{W})^{-1}\|_{\infty,2}^2)$. *If* $z = 1/w_{min}$ *and* $\sigma_{ub}^2 = \sigma_{max}^2 w_{max}$, *then for a fixed probability of error* $\delta \in (0, 1)$, *we have* $P(\mathit{TR}(G) = \hat{G}) \geq 1 - \delta$. *Where* $\hat{G} = (V, \hat{E})$ *is the learned graph by using Algorithm 1, and provided that* $m \in \mathcal{O}(\sigma_{ub}^2 \log \frac{n}{\delta})$ *interventional samples are used per* $\delta$-*noisy partially-correct path query in Algorithm 6.*

The values of $w_{min}$ and $w_{max}$ follow the specifications of Theorem 2. In addition, the value of $w_{min}$ is guaranteed to be greater than 0 because of the faithfulness assumption (see Assumption 3). For an example about our motivation to use the faithfulness assumption, see Appendix D.

## 3.3 Recovery of Transitive Edges

In this section, we show a method to recover the transitive edges by using multiple-vertex interventions. This allows us to learn the full network. For this purpose, we present a new query defined as follows.

**Definition 5** ($\delta$-noisy transitive query). *Let* $G = (V, E)$ *be a DAG, and let* $\delta \in (0, 1)$ *be a probability of error. A* $\delta$-*noisy transitive query is a function* $\tilde{T}_G : V \times V \times 2^V \to \{0, 1\}$ *such that* $\tilde{T}_G(i, j, S) = 1$ *with probability at least* $1 - \delta$ *if* $(i, j) \in E$ *is a transitive edge (where the additional path from* $i$ *to* $j$ *goes through* $S$), *and* 0 *otherwise. Here* $S \subseteq \pi_G(j)$ *is an auxiliary set necessary to answer the query, in order to block any influence from* $i$ *to* $S$, *and to unveil the direct effect from* $i$ *to* $j$.

Algorithms 7 and 8 (see Appendix B.3) show how to answer a transitive query for discrete and continuous CBNs respectively. Both algorithms are motivated on a property of CBNs, that is, $\forall i \in V$ and for every set $S$ disjoint of $\{i, \pi_G(i)\}$, we have $P(X_i | do(X_{\pi_G(i)} = x_{\pi_G(i)}), do(X_S = x_S)) = P(X_i | do(X_{\pi_G(i)} = x_{\pi_G(i)}))$. Thus, both algorithms intervene all the variables in $S$, if $S$ is the parent set of $j$, then $i$ will have no effect on $j$ and they return 0, and 1 otherwise.

Recall that by using Algorithm 1 we obtain the transitive reduction of the CBN, thus, we have the true topological ordering of the CBN, and also for each node $i \in V$, we know its parent set or a subset of it. Using these observations, we can cleverly set the input $i$, $j$, and $S$ of a noisy transitive query, as done in Algorithm 3. It is clear that Algorithm 3 makes $\mathcal{O}(n^2)$ noisy transitive queries in total. The time complexity to answer a transitive query for a discrete CBN is exponential in the maximum number of parents in the worst case. However, the sample complexity for queries in discrete and continuous CBNs remains polynomial in $n$ as prescribed in the following theorems.

**Theorem 3.** *Let* $\mathcal{B} = (G, \mathcal{P}_G)$ *be a discrete CBN, such that each random variable* $X_j$ *has a finite domain* $Dom[X_j]$, *with* $|Dom[X_j]| \leq r$. *Furthermore, let*

$$\gamma = \min_{\substack{j \in V \\ S \subseteq \pi_G(j), |S| \geq 1}} \min_{\substack{x_S, x_S' \in \times_{i \in S} Dom[X_i] \\ \mathbf{p}(X_j | do(X_S = x_S)) \neq \mathbf{p}(X_j | do(X_S = x_S'))}} \|\mathbf{p}(X_j | do(X_S = x_S)) - \mathbf{p}(X_j | do(X_S = x_S'))\|_\infty,$$

---
**Algorithm 3** Learning the transitive edges by using noisy transitive queries
---
**Input:** Transitively reduced DAG $\hat{G} = (V, \hat{E})$ (output of Algorithm 1)
**Output:** DAG $\tilde{G} = (V, \tilde{E})$
1: $\Psi \leftarrow \texttt{TopologicalOrder}(\hat{G}); \quad \hat{\pi}(i) \leftarrow \{u \in V | (u, i) \in \hat{E}\}$ (current parents of $i$); $\quad \tilde{E} \leftarrow \hat{E}$
2: **for** $j = 2 \dots n$ **do**
3:     **for** $i = j-1, j-2, \dots 1$ **do**
4:         **if** $\tilde{T}(\Psi_i, \Psi_j, \hat{\pi}(\Psi_j)) = 1$ **then** $\tilde{E} \leftarrow \tilde{E} \cup \{(\Psi_i, \Psi_j)\}$ and $\hat{\pi}(\Psi_j) \leftarrow \hat{\pi}(\Psi_j) \cup \Psi_i$
---

*and let $\tilde{G} = (V, \tilde{E})$ be the output of Algorithm 3. Then for $\gamma > 0$ and a fixed probability of error $\delta \in (0, 1)$, we have $P\left(G = \tilde{G}\right) \geq 1 - \delta$, provided that $m \in \mathcal{O}(\frac{1}{\gamma^2}\left(\ln n + \ln \frac{r}{\delta}\right))$ interventional samples are used per $\delta$-noisy transitive query in Algorithm 7.*

**Theorem 4.** *Let $\mathcal{B} = (G, \mathcal{P}_G)$ be a continuous CBN such that each variable $X_j$ is a sub-Gaussian random variable with full support on $\mathbb{R}$, with mean $\mu_j = 0$ and variance $\sigma_j^2$. Let $\mu_{j|do(X_S=\mathbf{1}z)}$ and $\sigma_{j|do(X_S=\mathbf{1}z)}^2$ denote the expected value and variance of $X_j$ after intervening each node of $X_S$ with value $z$. Furthermore, let*

$$\mu(\mathcal{B}, z_1, z_2) = \min_{j \in V, S \subseteq \pi_G(j), |S| \geq 2, i \in S} \left| \mu_{j|do(X_{S-\{i\}}=\mathbf{1}z_1, X_i=z_2)} \right|,$$

$$\sigma^2(\mathcal{B}, z_1, z_2) = \max\left( \max_{j \in V} \sigma_j^2, \max_{j \in V, S \subseteq \pi_G(j), |S| \geq 2, i \in S} \sigma_{j|do(X_{S-\{i\}}=\mathbf{1}z_1, X_i=z_2)}^2 \right),$$

*and let $\tilde{G} = (V, \tilde{E})$ be the output of Algorithm 3. If there exist an upper bound $\sigma_{ub}^2$ and finite values $z_1, z_2$ such that $\sigma^2(\mathcal{B}, z_1, z_2) \leq \sigma_{ub}^2$ and $\mu(\mathcal{B}, z_1, z_2) \geq 1$, then for a fixed probability of error $\delta \in (0, 1)$, we have $P\left(G = \tilde{G}\right) \geq 1 - \delta$, provided that $m \in \mathcal{O}(\sigma_{ub}^2 \log \frac{n}{\delta})$ interventional samples are used per $\delta$-noisy transitive query in Algorithm 8.*

Next, we show that ASGN networks can fulfill the conditions in Theorem 4.

**Corollary 2.** *Let $\mathcal{B} = (G, \mathcal{P}(\mathbf{W}, \mathcal{S}))$, and $\sigma_{max}^2$ follow the same definition as in Corollary 1. Let $w_{min} = \min_{ij} |W_{ij}|$, and $w_{max} = \max(\|(\mathbf{I} - \mathbf{W})^{-1}\|_{\infty,2}^2, \max_{j \in V, S \subseteq \pi_G(j)} \|(\mathbf{I} - \odot_S \mathbf{W})^{-1}\|_{\infty,2}^2)$. If $z_1 = 0, z_2 = 1/w_{min}$, and $\sigma_{ub}^2 = \sigma_{max}^2 w_{max}$, then for a fixed probability of error $\delta \in (0, 1)$, we have $P(G = \tilde{G}) \geq 1 - \delta$, provided that $m \in \mathcal{O}(\sigma_{ub}^2 \log \frac{n}{\delta})$ interventional samples are used per $\delta$-noisy transitive query in Algorithm 8.*

## 4   Extensions

**Learning rooted trees.** Here we make use of the results in [32], for rooted trees of node degree at most $d$. Theorem 4 in [32] states that for a fixed probability error $\delta \in (0, 1)$, one can reconstruct a rooted tree with probability $1 - \delta$ in $\mathcal{O}(\frac{1}{\delta} \frac{1}{(1/2-\epsilon)^2} dn \log^2 n \log \frac{dn}{\delta})$ time provided that a total of $\mathcal{O}(\frac{1}{(1/2-\epsilon)^2} n \log \frac{dn}{\delta})$ noisy path queries are used, where $\epsilon$ relates to the confidence of the noisy path query. The number of queries is improved with respect to the $n^2$ queries used for general DAGs in the previous section. Finally, recall that in the previous section we made use of *partially-correct* path queries, for this part we require a stronger version of noisy path query, which is defined below.

**Definition 6** ($\epsilon$-noisy path query). *Let $G = (V, E)$ be a DAG, and let $Q_G$ be a path query. Let $\epsilon \in (0, 1/2)$ be a probability of error. A $\epsilon$-noisy path query is a function $\tilde{Q}_G : V \times V \rightarrow \{0, 1\}$ such that $\tilde{Q}_G(i, j) = Q_G(i, j)$ with probability at least $1 - \epsilon$, and $\tilde{Q}_G(i, j) = 1 - Q_G(i, j)$ with probability at most $\epsilon$.*

The following states the sample complexity for exact learning of rooted trees in the discrete case.

**Proposition 3.** *Let $\mathcal{B} = (G, \mathcal{P}_G)$ be a discrete CBN, such that each random variable $X_j$ has a finite domain $Dom[X_j]$, with $\left|Dom[X_j]\right| \leq r$. Furthermore, let*

$$\gamma = \min_{\substack{j \in V \\ i \in V}} \min_{\substack{x_i, x_i' \in Dom[X_i] \\ \mathbf{p}(X_j|do(X_i=x_i)) \neq \mathbf{p}(X_j|do(X_i=x_i'))}} \| \mathbf{p}(X_j|do(X_i = x_i)) - \mathbf{p}(X_j|do(X_i = x_i')) \|_{\infty},$$

*and let* $\hat{G} = (V, \hat{E})$ *be the learned graph by using Algorithm 7 in [32]. Then for* $\gamma > 0$ *and a fixed probability of error* $\delta \in (0, 1)$*, we have* $P\left(G = \hat{G}\right) \geq 1 - \delta$*, provided that* $m \in \mathcal{O}(\frac{1}{\gamma^2}\left(\ln n + \ln \frac{r}{\delta}\right))$ *interventional samples are used per* $\delta$*-noisy path query in Algorithm 5.*

We use the same Algorithm 5 to answer a $\epsilon$-noisy path query. The difference is that now $\gamma$ represents the *minimum causal effect* among all pair nodes and not only parent-child nodes.

**On Imperfect Interventions.** Here we state some results on imperfect interventions. In Appendix C, we show that the sample complexity for discrete CBNs is scaled by $\alpha^{-1}$, where $\alpha$ accounts for the degree of uncertainty in the intervention. While for CBNs of sub-Gaussian random variables, the sample complexity still has the same dependence on an upper bound of the variances.

## 5 Experiments

In Appendix G.1, we tested our algorithms for perfect and imperfect interventions in synthetic networks, in order to empirically show the logarithmic phase transition of the number of interventional samples (see Figure 3 as an example). Appendix G.2 shows that in several benchmark BNs, most of the graph belongs to its transitive reduction, meaning that one can learn most of the network in polynomial time. Appendix G.3 shows experiments on some of these benchmark networks, using the aforementioned algorithms and also our algorithm for learning transitive edges, thus recovering the full networks. Finally, in Appendix G.4, as an illustration of the availability of interventional data, we show experimental evidence using three gene perturbation datasets from [33, 9].

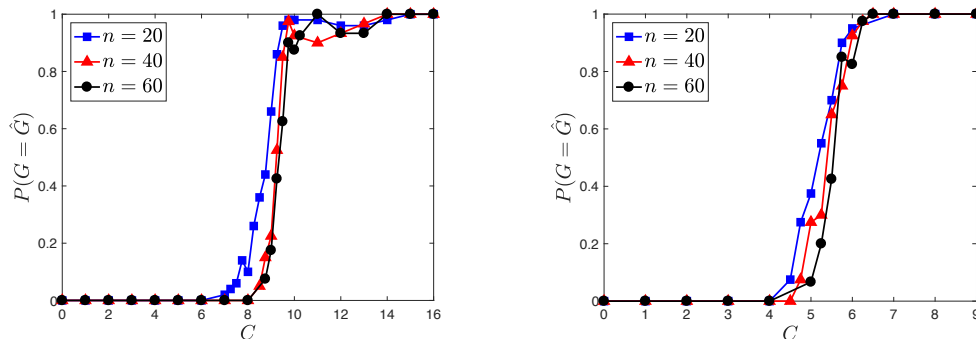

Figure 3: (Left) Probability of correct structure recovery of the transitive reduction of a discrete CBN vs. number of samples per query, where the latter was set to $e^C \log nr$, with all CBNs having $r = 5$ and $\gamma \geq 0.01$. (Right) Similarly, for continuous CBNs, the number of samples per query was set to $e^C \log n$, with all CBNs having $\|(\mathbf{I} - \mathbf{W})^{-1}\|_{2,\infty}^2 \leq 20$. Finally, we observe that there is a sharp phase transition from recovery failure to success in all cases, and the $\log n$ scaling holds in practice, as prescribed by Theorems 1, 2.

## 6 Future Work

There are several ways of extending this work. For instance, it would be interesting to analyze other classes of interventions with uncertainty, as in [7]. For continuous CBNs, we opted to use expected values and not to compare continuous distributions directly. The fact that the conditioning is with respect to a continuous random variable makes this task more complex than the typical comparison of continuous distributions. Still, it would be interesting to see whether kernel density estimators [16] could be beneficial.

## Footnotes

[1]Two graphs are Markov equivalent if they imply the same set of (conditional) independences. In general, two graphs are Markov equivalent iff they have the same structure ignoring arc directions, and have the same v-structures [31]. (A v-structure consists of converging directed edges into the same node, such as $X \to Y \leftarrow Z$.)

[2]This path query requires a "stronger" version of Definition 3. See for instance Definition 6.

[3]Several prior works from leading experts also have $\tilde{\mathcal{O}}(\frac{1}{\gamma^2})$ sample complexity for an *unknowable* constant $\gamma$. See for instance, [2, 20, 24].

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
