[Supplementary Material]

# SUPPLEMENTARY MATERIAL
# Computationally and statistically efficient learning of causal Bayes nets using path queries

## Appendix A   Discussion

**Learning causal Bayes nets from purely interventional data.**   Our interest in purely interventional data stems from our goal of discovering the true causal relationships. We perform single-vertex interventions for each node, which agrees with the numbers of single-vertex interventions sufficient and in the worst-case necessary to identify any DAG, as shown in [8].

**Availability of purely interventional data.**   The availability of purely interventional data is an implicit assumption in several prior works, which equivalently assume that one can perform an intervention on any node [19, 28, 11, 10, 25, 13]. As an illustration of the availability of interventional data, as well as the applicability of our method, we show experimental evidence using three gene perturbation datasets from [33, 9]. (See Appendix G.4.)

## Appendix B   Algorithms

### B.1   Algorithm for Transitive Reduction

As proved in [1], the time complexity of the best algorithm for finding the transitive reduction of a DAG is the same as the time to compute the transitive closure of a graph or to perform Boolean matrix multiplication. Therefore, we can use any exact algorithm for fast matrix multiplication, such as [15], which has $\mathcal{O}(n^{2.3729})$ time complexity. As a result, the time complexity of Algorithm 4 is dominated by the computation of the transitive reduction since answering a query $\tilde{Q}(i, j)$ is in $\tilde{\mathcal{O}}(\log n)$. Finally, note that performing $n^2$ queries (one per each node pair) is equivalent to performing $n$ single-vertex interventions, in which we intervene one node and observe the remaining $n - 1$ nodes. This number of interventions is necessary in the worst case, as discussed in [8].

---

**Algorithm 4** Learning the transitive reduction by using noisy path queries

**Input:** Vertex set V
**Output:** Edge set $\hat{\text{E}}$
 1: $\hat{\text{E}} \leftarrow \varnothing$
 2: **for** $i = 1 \ldots n$ **do**
 3:     **for** $j = 1 \ldots n$ **do**
 4:         **if** $i \neq j$ and $\tilde{Q}(i, j) = 1$ **then**
 5:             $\hat{\text{E}} \leftarrow \hat{\text{E}} \cup \{(i, j)\}$
 6: $\hat{\text{E}} \leftarrow \text{TR}(\hat{\text{E}})$

---

Assuming that we have correct answers for all path queries, Algorithm 4 will indeed exactly recover the $\text{TR}(\text{G})$ of any DAG G. However, this is not necessary. We can recover the true transitive reduction, $\text{TR}(\text{G})$, if we have correct answers for queries $Q_{\text{G}}(i, j)$ when $i \in \pi_{\text{G}}(j)$, and when there is no directed path from $i$ to $j$, and arbitrary answers when there is a directed path from $i$ to $j$. This is because the *transitive reduction* step will remove every transitive edge. It is the previous observation that motivated our characterization of noisy queries given in Definition 3.

### B.2   Noisy Path Query Algorithms

Algorithms 5 and 6 present our algorithms for answering a noisy path query $\tilde{Q}(i, j)$ motivated by Theorems 1 and 2 respectively. For discrete CBNs, we first create a list $\mathcal{L}$ of size $d = |Dom[X_i]|$,

containing the empirical probability mass functions (PMFs) of $X_j$ after intervening $X_i$ with all the possible values from its domain $Dom[X_i]$. Next, if the $\ell_\infty$-norm of the difference of any pair of PMFs in $\mathcal{L}$ is greater than a constant $\gamma$, then we answer the query with 1, and 0 otherwise. For continuous CBNs, we intervene $X_i$ with a constant value $z$ and compute the empirical expected value of $X_j$. We then output 1 if the absolute value of the expected value is greater than $1/2$, and 0 otherwise. (The threshold of $1/2$ is due to the particular way to set $z$, as prescribed by Theorem 2 and Corollary 1.)

---

**Algorithm 5** Noisy path query algorithm for discrete variables

---

**Input:** Nodes $i$ and $j$, number of interventional samples $m$, and constant $\gamma$.
**Output:** $\tilde{Q}(i,j)$
 1: $\mathcal{L} \leftarrow$ emptyList()
 2: **for** $x_i \in Dom[X_i]$ **do**
 3:      Intervene $X_i$ by setting its value to $x_i$, and obtain $m$ samples $x_j^{(1)}, \ldots, x_j^{(m)}$ of $X_j$
 4:      $\hat{p}_k = \frac{1}{m} \sum_{l=1}^{m} \mathbb{1}[x_j^{(l)} = k], \forall k \in Dom[X_j]$
 5:      Add $\hat{\mathbf{p}}$ to the list $\mathcal{L}$
 6: $\tilde{Q}(i,j) \leftarrow \mathbb{1}[(\exists\, \hat{\mathbf{p}}, \hat{\mathbf{q}} \in \mathcal{L})\, \|\hat{\mathbf{p}} - \hat{\mathbf{q}}\|_\infty > \gamma]$

---

**Algorithm 6** Noisy path query algorithm for continuous variables

---

**Input:** Nodes $i$ and $j$, number of interventional samples $m$, and constant $z$ (set as prescribed by Theorem 2 or Corollary 1.)
**Output:** $\tilde{Q}(i,j)$
 1: Intervene $X_i$ by setting its value to $z$, and obtain $m$ samples $x_j^{(1)}, \ldots, x_j^{(m)}$ of $X_j$
 2: $\hat{\mu} \leftarrow \frac{1}{m} \sum_{k=1}^{m} x_j^{(k)}$
 3: $\tilde{Q}(i,j) \leftarrow \mathbb{1}[|\hat{\mu}| > 1/2] \triangleright$ (The threshold of $1/2$ is due to the particular way to set $z$, as prescribed by Theorem 2 and Corollary 1.)

---

### B.3 Noisy Transitive Query Algorithms

Algorithms 7 and 8 show how to answer a transitive query for discrete and continuous CBNs respectively. Both algorithms are motivated on a property of CBNs, that is, $\forall i \in V$ and for every set S disjoint of $\{i, \pi_G(i)\}$, we have $P(X_i|do(X_{\pi_G(i)} = x_{\pi_G(i)}), do(X_S = x_S)) = P(X_i|do(X_{\pi_G(i)} = x_{\pi_G(i)}))$. Thus, both algorithms intervene all the variables in S, if S is the parent set of $j$, then $i$ will have no effect on $j$ and they return 0, and 1 otherwise.

---

**Algorithm 7** Noisy transitive query algorithm for discrete variables

---

**Input:** Nodes $i$ and $j$, set of nodes S, number of interventional samples $m$, and constant $\gamma$.
**Output:** $\tilde{T}(i,j,S)$
 1: $\mathcal{L} \leftarrow$ emptyList()
 2: **for** $x_s \in \times_{k \in S} Dom[X_k]$ **do**
 3:      Intervene set $X_S$ by setting its value to $x_s$
 4:      **for** $x_i \in Dom[X_i]$ **do**
 5:          Intervene $X_i$ by setting its value to $x_i$, and obtain $m$ samples $x_j^{(1)}, \ldots, x_j^{(m)}$ of $X_j$
 6:          $\hat{p}_k = \frac{1}{m} \sum_{l=1}^{m} \mathbb{1}[x_j^{(l)} = k], \forall k \in Dom[X_j]$
 7:          Add $\hat{\mathbf{p}}$ to the list $\mathcal{L}$
 8:      $\tilde{T}(i,j,S) \leftarrow \mathbb{1}[(\exists\, \hat{\mathbf{p}}, \hat{\mathbf{q}} \in \mathcal{L})\, \|\hat{\mathbf{p}} - \hat{\mathbf{q}}\|_\infty > \gamma]$
 9:      **if** $\tilde{T}(i,j,S) = 1$ **then** STOP

---

### B.4 Query Algorithm for Discrete Networks Under Imperfect Interventions

Algorithm 9 shows how to answer a noisy query for discrete CBNs under imperfect interventions.

---
**Algorithm 8** Noisy transitive query algorithm for continuous variables
---
**Input:** Nodes $i$ and $j$, set of nodes S, number of interventional samples $m$, and constants $z_1, z_2$ (set as prescribed by Theorem 4 or Corollary 2.)

**Output:** $\tilde{T}(i, j, \mathrm{S})$

  1: Intervene all variables $X_{\mathrm{S}}$ by setting their values to $z_1$
  2: Intervene $X_i$ by setting its value to $z_2$, and obtain $m$ samples $x_j^{(1)}, \ldots, x_j^{(m)}$ of $X_j$
  3: $\hat{\mu} \leftarrow \frac{1}{m} \sum_{k=1}^{m} x_j^{(k)}$
  4: $\tilde{T}(i, j, \mathrm{S}) \leftarrow \mathbb{1}[|\hat{\mu}| > \nicefrac{1}{2}]$ ▷ (The threshold of ½ is due to the particular way to set $z_1$ and $z_2$, as prescribed by Theorem 4 and Corollary 2.)
---

---
**Algorithm 9** Noisy path query algorithm for discrete variables under imperfect interventions.
---
**Input:** Nodes $i$ and $j$, number of interventional samples $m$, and constant $\gamma$

**Output:** $\tilde{Q}(i, j)$

  1: $\mathcal{L} \leftarrow$ emptyList()
  2: **for** $x_i \in Dom[X_i]$ **do**
  3:     Try to intervene $X_i$ with value $x_i$, and obtain $m$ pair samples $(x_i^{(1)}, x_j^{(1)}), \ldots, (x_i^{(m)}, x_j^{(m)})$ of $X_i$ and $X_j$
  4:     $\hat{p}_k = \frac{1}{\sum_{l=1}^{m} \mathbb{1}[x_i^{(l)} = x_i]} \sum_{l=1}^{m} \mathbb{1}[x_j^{(l)} = k \wedge x_i^{(l)} = x_i], \forall k \in Dom[X_j]$
  5:     Add $\hat{\mathbf{p}}$ to the list $\mathcal{L}$
  6: $\tilde{Q}(i, j) \leftarrow \mathbb{1}[(\exists\, \hat{\mathbf{p}}, \hat{\mathbf{q}} \in \mathcal{L})\, \|\hat{\mathbf{p}} - \hat{\mathbf{q}}\|_\infty > \gamma]$
---

## Appendix C   On Imperfect Interventions

In this section we relax the assumption of perfect interventions and analyze the sample complexity of a noisy path query. [7] analyzed a general framework of interventions named as *uncertain interventions*. In general terms, we model an imperfect intervention by adding some degree of uncertainty to the intervened variable. Note that the main distinction with respect to perfect interventions is that now the intervened variable is a random variable, meanwhile in perfect interventions the intervened variable is considered a constant.

**Discrete random variables.**   For a discrete CBN, we assume that an intervention follows a Bernoulli trial. That is, when one wants to intervene a variable $X_i$ with target value $v$, the probability that $X_i$ takes the target value $v$ is $\phi_i$, i.e., $P(X_i = v) = \phi_i$, and $P(X_i \neq v) = 1 - \phi_i$ otherwise.

To answer a noisy path query under this setting, we modify lines 3 and 4 of Algorithm 5. In line 3, we now get pair samples $\{(x_i^{(1)}, x_j^{(1)}), \ldots, (x_i^{(m)}, x_j^{(m)})\}$. In line 4, we know estimate $\mathbf{p}(X_j | do(X_i = x_i))$ as follows: $\forall k \in Dom[X_j], \hat{p}_k = \frac{1}{\sum_{l=1}^{m} \mathbb{1}[x_i^{(l)} = x_i]} \sum_{l=1}^{m} \mathbb{1}[x_j^{(l)} = k \wedge x_i^{(l)} = x_i]$. For completeness, we include the algorithm in Appendix B.4. Finally, the number of interventional samples $m$ is prescribed by the following theorem.

**Theorem 5.** *Let $\mathcal{B} = (\mathrm{G}, \mathcal{P}_{\mathrm{G}})$, $r$, and $\gamma$ follow the same definition as in Theorem 1. Let $\alpha$ be a constant such that for all $i \in \mathrm{V}$, $1/2 \leq \alpha \leq \phi_i$, in terms of imperfect interventions. Let $\hat{\mathrm{G}} = (\mathrm{V}, \hat{\mathrm{E}})$ be the output of Algorithm 4. Then for $\gamma > 0$ and a fixed probability of error $\delta \in (0, 1)$, we have $P(\mathit{TR}(\mathrm{G}) = \hat{\mathrm{G}}) \geq 1 - \delta$, provided that $m \in \mathcal{O}(\frac{1}{\alpha\gamma^2}(\ln n + \ln \frac{r}{\delta}))$ interventional samples are used per $\delta$-noisy partially-correct path query in the* modified *Algorithm 5 as described above.*

In practice, knowing the value of each $\phi_i$ can be hard to obtain, hence our motivation to introduce a lower bound $\alpha$ in Theorem 5.

**Continuous random variables.**   For continuous CBNs, we model an imperfect intervention by assuming that the intervened variable is also a sub-Gaussian variable. That is, when one intervenes a variable $X_i$ with target value $v$, $X_i$ becomes a sub-Gaussian variable with mean $v$ and variance $\nu_i^2$. Finally, we continue using Algorithm 6 to answer noisy path queries under this new setting.

**Theorem 6.** *Let $\mathcal{B} = (\mathrm{G}, \mathcal{P}_{\mathrm{G}})$, $\mu_j = 0$, and $\sigma_j^2$ follow the same definition as in Theorem 2. Let $\mu_{j|do(X_i=z)}$ and $\sigma_{j|do(X_i=z)}^2$ denote the expected value and variance of $X_j$ after perfectly intervening $X_i$ with value $z$. Furthermore, let $\mu(\mathcal{B}, z) = \min_{(i,j)\in\mathrm{E}} |\mathbb{E}_{X_i}[\mu_{j|do(X_i=z)}]|$, and $\sigma^2(\mathcal{B}, z) = \max(\max_{(i,j)\in\mathrm{E}} \mathbb{E}_{X_i}[\sigma_{j|do(X_i=z)}^2], \max_{j\in\mathrm{V}} \sigma_j^2)$. Let $\hat{\mathrm{G}} = (\mathrm{V}, \hat{\mathrm{E}})$ be the output of Algorithm 4. If there exist an upper bound $\sigma_{ub}^2$ and a finite value $z$ such that $\sigma^2(\mathcal{B}, z) \leq \sigma_{ub}^2$ and $\mu(\mathcal{B}, z) \geq 1$, then for a fixed probability of error $\delta \in (0, 1)$, we have $P(\mathtt{TR}(\mathrm{G}) = \hat{\mathrm{G}}) \geq 1 - \delta$, provided that $m \in \mathcal{O}(\sigma_{ub}^2 \log \frac{n}{\delta})$ interventional samples are used per $\delta$-noisy partially-correct path query in Algorithm 6.*

The motivation of the conditions in Theorem 6 are similar to Theorem 2. Next, we show that ASGN models can fulfill the conditions above.

**Corollary 3.** *Under the settings given in Corollary 1. If for all $j \in \mathrm{V}$, $\nu_j^2 \leq \sigma_{max}^2$ in terms of imperfect interventions. Then, for a fixed probability of error $\delta \in (0, 1)$, we have $P(\mathtt{TR}(\mathrm{G}) = \hat{\mathrm{G}}) \geq 1 - \delta$ provided that $m \in \mathcal{O}(\sigma_{ub}^2 \log \frac{n}{\delta})$ interventional samples are used per $\delta$-noisy partially-correct path query in Algorithm 6.*

## Appendix D   Examples

### D.1   Example for the use of faithfulness assumption

Consider the following ASGN network in Figure 4, assume that $X_1$ is intervened, then we have that the expected value of $X_3$ is 0 regardless of the value of the intervention. This occurs because the effect is canceled via the directed paths $\{(1, 2), (2, 3)\}$ and $\{(1, 3)\}$. This motivated us to use the faithfulness assumption and rule out such "pathological" parameterizations. Finally, in practice, the values of $w_{min}$ and $\sigma_{ub}^2$ are unknown. Fortunately, knowing a lower bound of $w_{min}$ and an upper bound of $\sigma_{ub}^2$ suffices for structure recovery.

Figure 4: An ASGN network in which the effect of $X_1$ on $X_3$ is none.

### D.2   Example about the Number of Experiments in the Worst Case for Multiple-Vertex Interventions

Consider the following DAG in Figure 5 of 6 binary variables. Such that, $P(A) = 0.5, P(B) = 0.2, P(C) = 0.1$. Let us also assume that $P(X_1|\neg A \neg B \neg C) = 0.4$, and for any other combination of $A, B, C$ we have $P(X_1|\cdot) = 0.8$.

Let us say that we perform a multiple-vertex intervention of $A, B, C$, and that we want to unveil the causal edge $(A, X_1)$. For this DAG we have that $P(X_1|do(A), do(B), do(C)) = P(X_1|ABC)$. Next let us say that we randomly select the configuration $A, \neg B, C$ for the intervention. Then $P(X_1|do(A\neg BC)) = P(X_1|A\neg BC) = 0.8$, in order to discover the causal edge, we also perform the following intervention, $P(X_1|do(\neg A\neg BC)) = P(X_1|\neg A\neg BC) = 0.8$. Which results in an "independence" or apparent no causal effect. In order to unveil the causal edge $(A, X_1)$, it is required to intervene with the configurations $A, \neg B, \neg C$ and $\neg A, \neg B, \neg C$, which in the worst case may be a single configuration out of an exponential number of possible configurations that allows to find the *direct* causal effect.

## Appendix E   On Latent Confounders

It is well-known that the existence of confounders imposes the most crucial problem for inferring causal relationships from observational data [17, 21]. However, since we perform single-vertex

Figure 5: DAG of 6 variables where we perform a multiple-vertex intervention.

interventions for every node in the CBN, the existence of hidden confounders does not impose a problem. In the leftmost graph of Figure 6, $X$ and $Y$ are associated observationally due to a hidden common cause, but neither of them is a cause of the other. By intervening $X$ or $Y$, we remove the "hidden edges". As a consequence, we are able to infer that neither $X$ nor $Y$ is a cause. The middle graph shows an association between $X$ and $Y$, and the need to intervene $X$ in order to discover that $X$ is a cause of $Y$. Finally, the rightmost graph shows that even in more complex latent configurations, by intervening $X$ we are removing any association between $X$ and $Y$ due to confounders.

Figure 6: Examples of a latent configurations that associate the variables $X$ and $Y$.

# Appendix F    Detailed Proofs

We now present the proofs of Propositions, Theorems and Corollaries from our main text.

## F.1    Proof of Proposition 1

*Proof.* The proof follows directly from rule 3 of do-calculus [21], which states that $P(X_j|do(X_i = x_i)) = P(X_j)$ if $(X_i \perp X_j)$ in the mutilated graph after the intervention on $X_i$. Since there is no directed path from $i$ to $j$, in the mutilated graph there is either no path or a path with a v-structure between $i$ and $j$, which implies the independence of $X_i$ and $X_j$.

For clarity, we also provide a longer (and equivalent) proof. The proof follows a d-separation argument. Let $\bar{\mathcal{B}}$ be the network after we perform an intervention on $X_i$ with value $x_i$, i.e., $\bar{\mathcal{B}}$ has the edge set $E \setminus \{(p_i, i) \mid p_i \in \pi_G(i)\}$. Let $anc_G(i)$ and $anc_G(j)$ be the ancestor set of $i$ and $j$ respectively. Now, if there is no directed path from $i$ to $j$ in $\mathcal{B}$ then there is no directed path in $\bar{\mathcal{B}}$ either, therefore, $i \notin anc_G(j)$. Also, $anc_G(i) = \varnothing$ as a consequence of intervening $X_i$. Next, we follow the d-separation procedure to determine if $X_i$ and $X_j$ are marginally independent in $\bar{\mathcal{B}}$. Since $anc_G(i) = \varnothing$, the ancestral graph of $i$ consists of just $i$ itself in isolation, moralizing and disorienting the edges of the ancestral graph of $j$ will not create a path from $i$ to $j$. Thus, guaranteeing the independence of $X_i$ and $X_j$, i.e., $P(X_j) = P(X_j|X_i)$ in $\bar{\mathcal{B}}$. Finally, since $P(X_j|\boldsymbol{X}_{\pi_G(j)})$ is fully specified by the parents of $j$ and these parents are not affected by $i$, we have that the marginal of $X_j$ in $\mathcal{B}$ remains unchanged in $\bar{\mathcal{B}}$, i.e., $P(X_j|do(X_i = x_i)) = P(X_j)$. $\square$

## F.2    Proof of Proposition 2

*Proof.* Here we assume faithfulness in the post-interventional distribution. Both claims follow a proof by contradiction. For Claim 1, if for all $x_i \in Dom[X_i]$ we have that $P(X_j) = P(X_j|do(X_i = x_i))$ then $X_i$ would not be a cause of $X_j$, which contradicts the fact that $i \in \pi_G(j)$. For Claim 2, if for all $x_i, x_i' \in Dom[X_i]$ we have that $P(X_j|do(X_i = x_i)) = P(X_j|do(X_i = x_i'))$ then in the mutilated graph we have that $P(X_j) = P(X_j|X_i = x_i)$ for all $x_i$, which implies that $X_i$ would not be a cause of $X_j$, thus contradicting the fact that $i \in \pi_G(j)$. $\square$

### F.3 Proof of Proposition 3

The proof follows similar arguments to the proof of Theorem 1.

### F.4 Proof of Theorem 1

To answer a path query in a discrete CBN, our algorithm compares two empirical PMFs, therefore, we need a good estimation of these PMFs. The following lemma shows the sample complexity to estimate several PMFs simultaneously by using maximum likelihood estimation.

**Lemma 1.** *Let $Y_1, \ldots, Y_L$ be $L$ random variables, such that w.l.o.g. the domain of each variable, $Dom[Y_i]$, is a finite subset of $\mathbb{Z}^+$. Also, let $y_i^{(1)}, \ldots, y_i^{(m)}$ be $m$ independent samples of $Y_i$. The maximum likelihood estimator, $\hat{\mathbf{p}}(Y_i)$, is obtained as follows:*

$$\hat{\mathrm{p}}_j(Y_i) = \frac{1}{m} \sum_{k=1}^{m} \mathbb{1}[y_i^{(k)} = j], \quad j \in Dom[Y_i].$$

*Then, for fixed values of $t > 0$ and $\delta \in (0,1)$, and provided that $m \geq \frac{2}{t^2} \ln \frac{2L}{\delta}$, we have*

$$P\left((\forall i \in \{1 \ldots L\}) \left\|\hat{\mathbf{p}}(Y_i) - \mathbf{p}(Y_i)\right\|_\infty \leq t\right) \geq 1 - \delta.$$

*Proof.* We use the Dvoretzky-Kiefer-Wolfowitz inequality [18, 6]:

$$P\left(\sup_{j \in Dom[Y_i]} \left|\hat{F}_j(Y_i) - F_j(Y_i)\right| > t\right) \leq 2e^{-2mt^2}, \quad t > 0,$$

where $\hat{F}_j(Y_i) = \sum_{k \leq j} \hat{\mathrm{p}}_k(Y_i)$ and $F_j(Y_i) = \sum_{k \leq j} \mathrm{p}_k(Y_i)$. Since $\hat{\mathrm{p}}_j(Y_i) = \hat{F}_j(Y_i) - \hat{F}_{j-1}(Y_i)$ and $\mathrm{p}_j(Y_i) = F_j(Y_i) - F_{j-1}(Y_i)$, we have

$$\left|\hat{\mathrm{p}}_j(Y_i) - \mathrm{p}_j(Y_i)\right| = \left|\left(\hat{F}_j(Y_i) - \hat{F}_{j-1}(Y_i)\right) - \left(F_j(Y_i) - F_{j-1}(Y_i)\right)\right|$$

$$\leq \left|\hat{F}_j(Y_i) - F_j(Y_i)\right| + \left|\hat{F}_{j-1}(Y_i) - F_{j-1}(Y_i)\right|$$

therefore, for a specific $i$, we have

$$P\left(\left\|\hat{\mathbf{p}}(Y_i) - \mathbf{p}(Y_i)\right\|_\infty > t\right) \leq 2e^{-mt^2/2}, \quad t > 0.$$

Then by the union bound, we have

$$P\left((\exists i \in \{1 \ldots L\}) \left\|\hat{\mathbf{p}}(Y_i) - \mathbf{p}(Y_i)\right\|_\infty > t\right) \leq 2Le^{-mt^2/2}, \quad t > 0.$$

Let $\delta = 2Le^{-mt^2/2}$, then for $m \geq \frac{2}{t^2} \ln \frac{2L}{\delta}$, we have

$$P\left((\forall i \in \{1 \ldots L\}) \left\|\hat{\mathbf{p}}(Y_i) - \mathbf{p}(Y_i)\right\|_\infty \leq t\right) \geq 1 - \delta, \quad \delta \in (0,1), \ t > 0.$$

Which concludes the proof of Lemma 1. $\qquad\qquad\square$

Lemma 1 states that simultaneously for all $L$ PMFs, the maximum likelihood estimator $\hat{\mathbf{p}}(Y_i)$ is at most $t$-away of $\mathbf{p}(Y_i)$ in $\ell_\infty$-norm with probability at least $1 - \delta$. Next, we provide the proof of Theorem 1.

*Proof.* We analyze a path query $\tilde{Q}(i,j)$ for nodes $i, j \in \mathrm{V}$. From the contrapositive of Proposition 1 we have that if $P(X_j|do(X_i = x_i)) \neq P(X_j)$ then there exists a directed path from $i$ to $j$. To detect the latter, we opt to use Claim 2 from Proposition 2.

Let $\mathbf{p}_{ij}^{(k)} = P(X_j|do(X_i = x_k))$ for all $i, j \in \mathrm{V}$ and $x_k \in Dom[X_i]$, and let $\hat{\mathbf{p}}_{ij}^{(k)}$ be the maximum likelihood estimation of $\mathbf{p}_{ij}^{(k)}$. Also, let $\tau = \frac{\gamma}{2}$ for convenience. Next, using Lemma 1 with $t = \tau/4$ and $L = rn^2$, we have

$$P\left((\forall i, j \in \mathrm{V}, \forall x_k \in Dom[X_i]) \left\|\hat{\mathbf{p}}_{ij}^{(k)} - \mathbf{p}_{ij}^{(k)}\right\|_\infty \leq \tau/4\right) \geq 1 - \delta.$$

That is, with probability at least $1 - \delta$, simultaneously for all $i, j, k$, the estimators $\hat{\mathbf{p}}_{ij}^{(k)}$ are at most $\tau/4$-away from the true distributions $\mathbf{p}_{ij}^{(k)}$ in $\ell_\infty$ norm, provided that $m \geq \frac{32}{\tau^2}(2 \ln n + \ln \frac{2r}{\delta})$ samples are used in the estimation.

Now, we analyze the two cases that we are interested to answer with high probability. First, let $i \in \pi_{\mathrm{G}}(j)$. We have that for any two distributions $\mathbf{p}_{ij}^{(u)}, \mathbf{p}_{ij}^{(v)}$ where $x_u, x_v \in Dom[X_i]$, either $\mathbf{p}_{ij}^{(u)} = \mathbf{p}_{ij}^{(v)}$ or $\|\mathbf{p}_{ij}^{(u)} - \mathbf{p}_{ij}^{(v)}\|_\infty > \tau$ (recall the definition of $\gamma$ and $\tau$). Next, for a specific $i, j$, we show how to test if two distributions $\mathbf{p}_{ij}^{(u)}, \mathbf{p}_{ij}^{(v)}$ are equal or not. Let us assume $\mathbf{p}_{ij}^{(u)} = \mathbf{p}_{ij}^{(v)}$, then we have

$$
\begin{aligned}
\left\|\hat{\mathbf{p}}_{ij}^{(u)} - \hat{\mathbf{p}}_{ij}^{(v)}\right\|_\infty &= \left\|\hat{\mathbf{p}}_{ij}^{(u)} - \mathbf{p}_{ij}^{(u)} - \left(\hat{\mathbf{p}}_{ij}^{(v)} - \mathbf{p}_{ij}^{(v)}\right)\right\|_\infty \\
&\leq \left\|\hat{\mathbf{p}}_{ij}^{(u)} - \mathbf{p}_{ij}^{(u)}\right\|_\infty + \left\|\hat{\mathbf{p}}_{ij}^{(v)} - \mathbf{p}_{ij}^{(v)}\right\|_\infty \\
&\leq \tau/2.
\end{aligned}
$$

Therefore, if $\|\hat{\mathbf{p}}_{ij}^{(u)} - \hat{\mathbf{p}}_{ij}^{(v)}\|_\infty > \tau/2$ then w.h.p. $\mathbf{p}_{ij}^{(u)} \neq \mathbf{p}_{ij}^{(v)}$. On the other hand, if $\|\hat{\mathbf{p}}_{ij}^{(u)} - \hat{\mathbf{p}}_{ij}^{(v)}\|_\infty \leq \tau/2$ then w.h.p. we have:

$$
\begin{aligned}
\left\|\mathbf{p}_{ij}^{(u)} - \mathbf{p}_{ij}^{(v)}\right\|_\infty &= \left\|\mathbf{p}_{ij}^{(u)} - \hat{\mathbf{p}}_{ij}^{(u)} - \left(\mathbf{p}_{ij}^{(v)} - \hat{\mathbf{p}}_{ij}^{(v)}\right) + \hat{\mathbf{p}}_{ij}^{(u)} - \hat{\mathbf{p}}_{ij}^{(v)}\right\|_\infty \\
&\leq \left\|\hat{\mathbf{p}}_{ij}^{(u)} - \mathbf{p}_{ij}^{(u)}\right\|_\infty + \left\|\hat{\mathbf{p}}_{ij}^{(v)} - \mathbf{p}_{ij}^{(v)}\right\|_\infty + \left\|\hat{\mathbf{p}}_{ij}^{(u)} - \hat{\mathbf{p}}_{ij}^{(v)}\right\|_\infty \\
&\leq \tau.
\end{aligned}
$$

From the definition of $\gamma$ and $\tau$, we have $\|\mathbf{p}_{ij}^{(u)} - \mathbf{p}_{ij}^{(v)}\|_\infty > \tau$ for any pair $\mathbf{p}_{ij}^{(u)} \neq \mathbf{p}_{ij}^{(v)}$, then w.h.p. we have that $\mathbf{p}_{ij}^{(u)} = \mathbf{p}_{ij}^{(v)}$.

Second, let be the case that there is no directed path from $i$ to $j$. Then, following Proposition 1, we have that all the distributions $\mathbf{p}_{ij}^{(k)}, \forall x_k \in Dom[X_i]$, are equal. Similarly as in the first case, we have that if $\|\hat{\mathbf{p}}_{ij}^{(u)} - \hat{\mathbf{p}}_{ij}^{(v)}\|_\infty > \tau/2$ then w.h.p. $\mathbf{p}_{ij}^{(u)} \neq \mathbf{p}_{ij}^{(v)}$, and equal otherwise.

Next, note that since Algorithm 5 compares pair of distributions, the provable guarantee of *all* queries (after eliminating the transitive edges) is directly related to the estimation of *all* PMFs with probability of error at most $\delta$, i.e., we have that

$$
P\left(\left(\forall j = 1, \ldots, n \wedge (i \in \pi_{\mathrm{G}}(j) \vee j \notin desc_{\mathrm{G}}(i))\right) \tilde{Q}(i,j) = Q_{\mathrm{G}}(i,j)\right) \geq 1 - \delta,
$$

where $desc_{\mathrm{G}}(i)$ denotes the descendants of $i$. Finally, note that we are estimating each distribution by using $m \geq \frac{32}{\tau^2}(2 \ln n + \ln \frac{2r}{\delta})$ samples, i.e., $m \in \mathcal{O}(\frac{1}{\gamma^2}(\ln n + \ln \frac{r}{\delta}))$. However, for each query $\tilde{Q}(i,j)$ in Algorithm 5, we estimate a maximum of $r$ distributions, as a result, we use $\frac{32r}{\tau^2}(2 \ln n + \ln \frac{2r}{\delta})$ interventional samples in total per query. $\qquad\square$

### F.5 Proof of Theorem 2

*Proof.* From the contrapositive of Proposition 1 we have that if $P(X_j|do(X_i = x_i)) \neq P(X_j)$ then there exists a directed path from $i$ to $j$. To detect the latter, we opt to use Claim 1 from Proposition 2, i.e., using expected values. Recall from the characterization of the BN that there exist a finite value $z$ and upper bound $\sigma_{ub}^2$, such that $\mu(\mathcal{B}, z) \geq 1$ and $\sigma^2(\mathcal{B}, z) \leq \sigma_{ub}^2$. Let $x_j^{(1)}, \ldots, x_j^{(m)}$ be $m$ i.i.d. samples of $X_j$ after intervening $X_i$ with $z$, and let $\mu_{j|do(X_i=z)}$ and $\sigma_{j|do(X_i=z)}^2$ be the mean and variance of $X_j$ respectively. Also, let $\hat{\mu}_{j|do(X_i=z)} = \frac{1}{m}\sum_{k=1}^m x_j^{(k)}$ be the empirical expected value of $X_j$.

Now, we analyze the two cases that we are interested to answer with high probability. First, let $i \in \pi_{\mathrm{G}}(j)$. Clearly, $\hat{\mu}_{j|do(X_i=z)}$ has expected value $|\mathbb{E}[\hat{\mu}_{j|do(X_i=z)}]| = |\mu_{j|do(X_i=z)}| \geq 1$, and variance $\hat{\sigma}_{j|do(X_i=z)}^2 = \sigma_{j|do(X_i=z)}^2/m \leq \sigma_{ub}^2/m$. Then, using Hoeffding's inequality we have

$$
P\left(\left|\hat{\mu}_{j|do(X_i=z)} - \mu_{j|do(X_i=z)}\right| \geq t\right) \leq 2e^{-t^2/(2\hat{\sigma}_{j|do(X_i=z)}^2)}
$$

$$\leq 2e^{-mt^2/(2\sigma_{ub}^2)}. \tag{6.1}$$

Second, if there is no directed path from $i$ to $j$, then by using Proposition 1, we have $\mu_{j|do(X_i=z)} = \mu_j = 0$ and $\sigma_{j|do(X_i=z)}^2 = \sigma_j^2 \leq \sigma_{ub}^2$.

As we can observe from both cases described above, the true mean $\mu_{j|do(X_i=z)}$ when $i \in \pi_G(j)$ is at least separated by 1 from the true mean when there is no directed path. Therefore, to estimate the mean, a suitable value for $t$ in inequality (6.1) is $t \leq 1/2$. The latter allows us to state that if $|\hat{\mu}_{j|do(X_i=z)}| > 1/2$ then $\tilde{Q}(i,j) = 1$, and $\tilde{Q}(i,j) = 0$ otherwise. Replacing $t = 1/2$ and restating inequality (6.1), we have that for a specific pair of nodes $(i,j)$, if $i \in \pi_G(j)$ or if $j \notin desc_G(i)$ ($desc_G(i)$ denotes the descendants of $i$), then

$$P\left(Q_G(i,j) \neq \tilde{Q}(i,j)\right) \leq 2e^{-m/(8\sigma_{ub}^2)}.$$

The latter inequality is for a single query. Using the union bound we have

$$P\left(\left(\exists j = 1,\ldots,n \wedge (i \in \pi_G(j) \vee j \notin desc_G(i))\right) \tilde{Q}(i,j) \neq Q_G(i,j)\right) \leq 2n^2 e^{-m/(8\sigma_{ub}^2)}.$$

Now, let $\delta = 2n^2 e^{-m/(8\sigma_{ub}^2)}$, if $m \geq 8\sigma_{ub}^2 \log \frac{2n^2}{\delta}$ then

$$P\left(\left(\forall j = 1,\ldots,n \wedge (i \in \pi_G(j) \vee j \notin desc_G(i))\right) \tilde{Q}(i,j) = Q_G(i,j)\right) \geq 1 - \delta.$$

That is, with probability of at least $1 - \delta$, the path query $\tilde{Q}(i,j)$ (in Algorithm 6) is equal to $Q_G(i,j)$ for all $n^2$ performed queries in which either $i \in \pi_G(j)$, or there is no directed path from $i$ to $j$. Note also that the probability at least $1 - \delta$ is guaranteed after we remove the transitive edges in the network. Therefore, we obtain $m \geq 8\sigma_{ub}^2(2\log n + \log \frac{2}{\delta})$, i.e., $m \in \mathcal{O}(\sigma_{ub}^2 \log \frac{n}{\delta})$. □

## F.6 Proof of Theorem 3

The proof follows the same arguments given in the proof of Theorem 1. For a pair of nodes $i, j$, Algorithm 3 sets $S = \hat{\pi}_G(j)$. If S is already the true parent set of $j$, then $X_i$ will only have effect on $X_j$ if $i \in S$. If S is a subset of the true parent set, then $X_i$ will only have effect on $X_j$ if there exists a transitive edge $(i,j)$. This is because by intervening S we are blocking any possible effect of $X_i$ on $X_j$ through any node in S, and since non-transitive edges are already recovered then $(i,j)$ must be a transitive edge if there exists some effect. This effect is detected as in Theorem 1, i.e., through the $\ell_\infty$-norm of difference of empirical marginals of $X_j$.

## F.7 Proof of Theorem 4

The proof follows the same arguments given in the proof of Theorem 2. For a pair of nodes $i, j$, Algorithm 3 sets $S = \hat{\pi}_G(j)$. If S is already the true parent set of $j$, then $X_i$ will only have effect on $X_j$ if $i \in S$. If S is a subset of the true parent set, then $X_i$ will only have effect on $X_j$ if there exists a transitive edge $(i,j)$. This is because by intervening S we are blocking any possible effect of $X_i$ on $X_j$ through any node in S, and since non-transitive edges are already recovered then $(i,j)$ must be a transitive edge if there exists some effect. This effect is detected as in Theorem 2, i.e., through the absolute value of the difference of the empirical means of $X_j$.

## F.8 Proof of Theorem 5

To prove Theorem 5 we first derive a lemma that specifies the number of samples to obtain a good approximation with guarantees of conditional PMFs.

**Lemma 2.** *Let $Y_1, \ldots, Y_L$ be L discrete random variables, such that w.l.o.g. the domain of each variable, $Dom[Y_i]$, is a finite subset of $\mathbb{Z}^+$. Let $Z_1, \ldots, Z_L$ be L Bernoulli random variables, such that each variable fulfills $P(Z_i = 1) \geq \alpha \geq 1/2$. Also, let $(z_i^{(1)}, y_i^{(1)}), \ldots, (z_i^{(m)}, y_i^{(m)})$ be m pair of independent samples of $Z_i$ and $Y_i$. The conditional maximum likelihood estimator, $\hat{\mathbf{p}}(Y_i|Z_i = 1)$, is obtained as follows:*

$$\hat{p}_j(Y_i|Z_i = 1) = \frac{1}{\sum_{k=1}^m z_i^{(k)}} \sum_{k=1}^m \mathbb{1}[y_i^{(k)} = j \wedge z_i^{(k)}], \quad j \in Dom[Y_i].$$

*Then, for fixed values of $t, \delta \in (0, 1)$, and provided that $m \geq \frac{4}{\alpha t^2} \ln \frac{4L}{\delta}$, we have*

$$P\left((\forall i \in \{1 \dots L\}) \left\|\hat{\mathbf{p}}(Y_i|Z_i = 1) - \mathbf{p}(Y_i|Z_i = 1)\right\|_\infty \leq t\right) \geq 1 - \delta.$$

*Proof.* First, we analyze a pair of variables $Z_i, Y_i$. Let $\mathcal{E}_1 = \{\frac{1}{m} \sum_{k=1}^m z_i^{(k)} \geq \alpha - \epsilon\}$. Next, using the one-sided Hoeffding's inequality, we have

$$P(\mathcal{E}_1) \geq 1 - e^{-2\epsilon^2 m}.$$

Now, let the event $\mathcal{E}_2 = \{\|\hat{\mathbf{p}}(Y_i|Z_i = 1) - \mathbf{p}(Y_i|Z_i = 1)\|_\infty \leq t\}$. Using Lemma 1 (see Proof F.4), we obtain

$$P(\mathcal{E}_2|\mathcal{E}_1) \geq 1 - 2e^{-m(\alpha - \epsilon)t^2/2}.$$

Then, by the law of total probability, we have

$$\begin{aligned} P(\mathcal{E}_2) &\geq P(\mathcal{E}_2|\mathcal{E}_1)P(\mathcal{E}_1) \\ &\geq 1 - e^{-2\epsilon^2 m} - 2e^{-m(\alpha - \epsilon)t^2/2}. \end{aligned}$$

Let $\frac{\delta}{2} = e^{-2\epsilon^2 m}$, and $\frac{\delta}{2} = 2e^{-m(\alpha - \epsilon)t^2/2}$. Then provided that $m \geq \max(\frac{1}{2\epsilon^2} \ln \frac{2}{\delta}, \frac{2}{(\alpha - \epsilon)t^2} \ln \frac{4}{\delta})$,

$$P(\mathcal{E}_2) \geq 1 - \delta.$$

For $\epsilon = \frac{\alpha}{2}$, and $t \in (0, 1)$, we can simplify the bound on $m$ to be $m \geq \frac{4}{\alpha t^2} \ln \frac{4}{\delta}$. Finally, using union bound and provided that $m \geq \frac{4}{\alpha t^2} \ln \frac{4L}{\delta}$, we have

$$P\left((\forall i \in \{1 \dots L\}) \left\|\hat{\mathbf{p}}(Y_i|Z_i = 1) - \mathbf{p}(Y_i|Z_i = 1)\right\|_\infty \leq t\right) \geq 1 - \delta.$$

Which concludes the proof. $\qquad\square$

Now follows the proof of Theorem 5.

*Proof of Theorem 5.* The proof follows the same steps as in the proof of Theorem 1 (Appendix F.4). The difference is that we now use the sample complexity given by Lemma 2 instead of Lemma 1. Therefore, for a query $\tilde{Q}(i, j)$ we obtain a sample complexity of $m \in \mathcal{O}(\frac{1}{\alpha \gamma^2}(\ln n + \ln \frac{r}{\delta}))$. $\qquad\square$

## F.9 Proof of Theorem 6

*Proof.* Recall from the characterization of the BN that there exist a finite value $z$ and upper bound $\sigma_{ub}^2$, such that $\mu(\mathcal{B}, z) \geq 1$ and $\sigma^2(\mathcal{B}, z) \leq \sigma_{ub}^2$. Let $x_j^{(1)}, \dots, x_j^{(m)}$ be $m$ i.i.d. samples of $X_j$ after trying to intervene $X_i$ with value $z$. Let $\mu_{j|do(X_i=z)}$ and $\sigma_{j|do(X_i=z)}^2$ be the mean and variance of $X_j$ respectively, after perfectly intervening $X_i$ with value $z$. Also, let $\hat{\mu} = \frac{1}{m} \sum_{k=1}^m x_j^{(k)}$ be the empirical expected value of $X_j$.

Now, we analyze the two cases that we are interested to answer with high probability. First, let $i \in \pi_G(j)$. Clearly, $\hat{\mu}$ has expected value $|\mathbb{E}[\hat{\mu}]| = |\mathbb{E}_{X_i}[\mu_{j|do(X_i=z)}]| \geq 1$, and variance $\hat{\sigma}^2 = \mathbb{E}_{X_i}[\sigma_{j|do(X_i=z)}^2]/m \leq \sigma_{ub}^2/m$. Then, using Hoeffding's inequality we have

$$\begin{aligned} P\left(\left|\hat{\mu} - \mathbb{E}[\hat{\mu}]\right| \geq t\right) &\leq 2e^{-t^2/(2\hat{\sigma}^2)} \\ &\leq 2e^{-mt^2/(2\sigma_{ub}^2)}. \end{aligned} \tag{6.2}$$

Second, if there is no directed path from $i$ to $j$, then by using Proposition 1, we have $\mathbb{E}_{X_i}[\mu_{j|do(X_i=z)}] = \mathbb{E}_{X_i}[\mu_j] = 0$ and $\mathbb{E}_{X_i}[\sigma_{j|do(X_i=z)}^2] = \mathbb{E}_{X_i}[\sigma_j^2] \leq \sigma_{ub}^2$.

As we can observe from both cases described above, the true mean $\mathbb{E}_{X_i}[\mu_{j|do(X_i=z)}]$ when $i \in \pi_G(j)$ is at least separated by 1 from the true mean when there is no directed path. Therefore, to estimate the mean, a suitable value for $t$ in inequality (6.2) is $t \leq 1/2$. The latter allows us to state that if $|\hat{\mu}| > 1/2$ then $\tilde{Q}(i, j) = 1$, and $\tilde{Q}(i, j) = 0$ otherwise. Replacing $t = 1/2$ and restating inequality

(6.2), we have that for a specific pair of nodes $(i, j)$, if $i \in \pi_{\mathrm{G}}(j)$ or if $j \notin desc_{\mathrm{G}}(i)$ ($desc_{\mathrm{G}}(i)$ denotes the descendants of $i$), then

$$P\left(Q_{\mathrm{G}}(i,j) \neq \tilde{Q}(i,j)\right) \leq 2e^{-m/(8\sigma_{ub}^2)}.$$

The latter inequality is for a single query. Using the union bound we have

$$P\left(\left(\exists j = 1, \ldots, n \wedge (i \in \pi_{\mathrm{G}}(j) \vee j \notin desc_{\mathrm{G}}(i))\right) \tilde{Q}(i,j) \neq Q_{\mathrm{G}}(i,j)\right) \leq 2n^2 e^{-m/(8\sigma_{ub}^2)}.$$

Now, let $\delta = 2n^2 e^{-m/(8\sigma_{ub}^2)}$, if $m \geq 8\sigma_{ub}^2 \log \frac{2n^2}{\delta}$ then

$$P\left(\left(\forall j = 1, \ldots, n \wedge (i \in \pi_{\mathrm{G}}(j) \vee j \notin desc_{\mathrm{G}}(i))\right) \tilde{Q}(i,j) = Q_{\mathrm{G}}(i,j)\right) \geq 1 - \delta.$$

That is, with probability of at least $1 - \delta$, the path query $\tilde{Q}(i,j)$ (in Algorithm 6) is equal to $Q_{\mathrm{G}}(i,j)$ for all $n^2$ performed queries in which either $i \in \pi_{\mathrm{G}}(j)$, or there is no directed path from $i$ to $j$. Note also that the probability at least $1 - \delta$ is guaranteed after we remove the transitive edges in the network. Therefore, we obtain $m \geq 8\sigma_{ub}^2(2\log n + \log \frac{2}{\delta})$, i.e., $m \in \mathcal{O}(\sigma_{ub}^2 \log \frac{n}{\delta})$. □

### F.10 Proof of Corollary 1

*Proof.* Let us first analyze the expected value $\mu_j$ of each variable $X_j$ in the network before performing any intervention. From the definition of the ASGN model we have that the expected value of $X_j$ is $\mu_j = \sum_{p \in \pi_{\mathrm{G}}(j)} W_{jp}\mu_p$, and from the topological ordering of the network we can observe that the variables without parents have zero mean since these are only affected by a sub-Gaussian noise with zero mean. Therefore, following this ordering we have that the mean of every variable $X_j$ is $\mu_j = 0$.

Recall from Remark 2 that we can write the model as: $X = \mathbf{W}X + N$, which is equivalent to $X = (\mathbf{I} - \mathbf{W})^{-1}N$. Let $\mathbf{B} = (\mathbf{I} - \mathbf{W})^{-1}$, then $\mathbf{B}_{ji}$ denotes the total *weight effect* of the noise $N_i$ on the node $j$. Furthermore, let $\odot_i \mathbf{B} = (\mathbf{I} - \odot_i \mathbf{W})^{-1}$ and similarly $\{\odot_i \mathbf{B}\}_{jk}$ denotes the total weight effect of the noise $N_k$ on the node $j$ after intervening the node $i$.

Next, we analyze if $z = 1/w_{min}$, and $\sigma_{ub}^2 = \sigma_{max}^2 w_{max}$ fulfill the conditions given in Theorem 2. First, let $i \in \pi_{\mathrm{G}}(j)$, i.e., $(i,j) \in \mathrm{E}$. Since $w_{min} = \min_{(i,j) \in \mathrm{E}} |\{\odot_i \mathbf{B}\}_{ji}|$, we have $|\mu_{j|do(X_i=z)}| = |\{\odot_i \mathbf{B}\}_{ji}| \times |z| = |\{\odot_i \mathbf{B}\}_{ji}|/w_{min}$. Since $w_{min} \leq |\{\odot_i \mathbf{B}\}_{ji}|$ for any $(i,j) \in \mathrm{E}$, we have that $\mu(\mathcal{B}, z) \geq 1$. Let $v_{j|do(X_i=z)}$ be the variance of $X_j$ after intervening $X_i$, then we have that $v_{j|do(X_i=z)}^2 = \sum_{p \in \mathrm{V} \backslash i}(\{\odot_i \mathbf{B}\}_{jp})^2 \sigma_j^2$, similarly the variance of $j$ without any intervention is $v_j^2 = \sum_{p \in \mathrm{V} \backslash i}(\mathbf{B}_{jp})^2 \sigma_j^2$. Then $\max_{(i,j) \in \mathrm{E}} v_{j|do(X_i=z)}^2 \leq \max_{i \in \mathrm{V}} \sigma_{max}^2 \|\odot_i \mathbf{B}\|_{\infty,2}^2$, and $\max_{j \in \mathrm{V}} v_j^2 \leq \sigma_{max}^2 \|\mathbf{B}\|_{\infty,2}^2$, which results in $\sigma_{ub}^2 = \sigma_{max}^2 w_{max}$.

Second, let be the case that there is no directed path from $i$ to $j$. Then from Proposition 1, $X_i$ and $X_j$ are independent after intervening $X_i$, i.e., $\mu_{j|do(X_i=z)} = \mu_j = 0$, and $v_{j|do(X_i=z)}^2 = v_j^2 \leq \sigma_{ub}^2$.

As shown above, for these values of $z = 1/w_{min}$ and $\sigma_{ub}^2 = \sigma_{max}^2 w_{max}$, we fulfill the conditions given in Theorem 2, which concludes our proof. □

### F.11 Proof of Corollary 2

For a pair of nodes $i, j$, Algorithm 3 sets $\mathrm{S} = \hat{\pi}_{\mathrm{G}}(j)$. If S is already the true parent set of $j$, then $X_i$ will only have effect on $X_j$ if $i \in \mathrm{S}$. If S is a subset of the true parent set, then $X_i$ will only have effect on $X_j$ if there exists a transitive edge $(i, j)$. This is because by intervening S we are blocking any possible effect of $X_i$ on $X_j$ through any node in S, and since non-transitive edges are already recovered then $(i, j)$ must be a transitive edge if there exists some effect. Thus, $w_{min} = \min_{ij} |W_{ij}|$ is enough to ensure a mean of at least 1 for $X_j$, since only $X_i$ is intervened with value $z_2 = 1/w_{min}$ while the other nodes in S are intervened with value $z_1 = 0$. Finally, because the value of $w_{max}$ takes the maximum across all possible interventions of subsets of the parent set of $j$, then $\sigma_{ub}^2$ is an upper bound and similar arguments as in Corollary 1 hold.

### F.12 Proof of Corollary 3

*Proof.* To prove the corollary we need to show that for $z = 1/w_{min}$ and $\sigma_{ub}^2 = \sigma_{max}^2 w_{max}$, the conditions $\mu(\mathcal{B}, z) \geq 1$ and $\sigma^2(\mathcal{B}, z) \leq \sigma_{ub}^2$ hold, similarly to Proof F.10.

For the case when $i \in \pi_{\mathrm{G}}(j)$, now $X_i$ (the intervened variable) is a sub-Gaussian variable with mean $z$ and variance $\nu_i^2$, we clearly have that the same upper bound $\sigma_{ub} = \sigma_{max}^2 w_{max}$ works since $\nu_i^2 \leq \sigma_{max}^2$. Likewise, the value $z$ is properly set since the value of $w_{min}$ is $w_{min} = \min_{(i,j) \in \mathrm{E}} |\{\odot_i \mathbf{B}\}_{ji}|$.

For the case when there is no directed path from $i$ to $j$, we have that $X_i$ and $X_j$ are independent after intervening $X_i$, i.e., $\mathbb{E}[X_j] = \mu_j = 0$, and $\mathrm{Var}[X_j] = \upsilon_j^2 \leq \sigma_{ub}^2$.

From these analyses we conclude that the ASGN model fulfills the conditions given in Theorem 6. Which concludes our proof. $\qquad\square$

## Appendix G    Experiments

### G.1    Experiments on Synthetic CBNs

In this section, we validate our theoretical results on synthetic data for perfect and imperfect interventions by using Algorithms 4, 5, and 6. Our objective is to characterize the number of interventional samples per query needed by our algorithm for learning the transitive reduction of a CBN exactly.

Our experimental setup is as follows. We sample a random transitively reduced DAG structure G over $n$ nodes. We then generate a CBN as follows: for a discrete CBN, the domain of a variable $X_i$ is $Dom[X_i] = \{1, \ldots, d\}$, where $d$ is the size of the domain, which is selected uniformly at random from $\{2, \ldots, 5\}$, i.e., $r = 5$ in terms of Theorem 1. Then, each row of a CPT is generated uniformly at random. Finally, we ensure that the generated CBN fulfills $\gamma \geq 0.01$. For a continuous CBN, we use Gaussian noises following the ASGN model as described in Definition 4, where each noise variable $N_i$ is Gaussian with mean 0 and variance selected uniformly at random from $[1, 5]$, i.e., $\sigma_{max}^2 = 5$, in terms of Corollary 1. The edge weights $W_{ij}$ are selected uniformly at random from $[-1.25, -0.01] \cup [0.01, 1.25]$ for all $(i, j) \in \mathrm{E}$. We ensure that $\mathbf{W}$ fulfills $\|(\mathbf{I} - \mathbf{W})^{-1}\|_{2,\infty}^2 \leq 20$. After generating a CBN, one can now intervene a variable, and sample accordingly to a given query. Finally, we set $\delta = 0.01$, and estimate the probability $P(\mathrm{G} = \hat{\mathrm{G}})$ by computing the fraction of times that the learned DAG structure $\hat{\mathrm{G}}$ matched the true DAG structure G exactly, across 40 randomly sampled BNs. We repeated this process for $n \in \{20, 40, 60\}$. The number of samples per query was set to $e^C \log nr$ for discrete BNs, and $e^C \log n$ for continuous BNs, where $C$ was the control parameter, chosen to be in $[0, 16]$. Figure 7 shows the results of the structure learning experiments. We can observe that there is a sharp phase transition from recovery failure to success in all cases, and that the $\log n$ scaling holds in practice, as prescribed by Theorems 1 and 2.

Similarly, for imperfect interventions we work under the same experimental settings described above. For a discrete BN, we additionally set $\alpha = 0.9$ in terms of Theorem 5. Whereas for a continuous BN, we set $\nu_i^2 = \sigma_i^2$ for all $i \in \mathrm{V}$, in terms of 3. Figure 7 shows the results of the structure learning experiments. We can observe that the sharp phase transition from recovery failure to success and the $\log n$ scaling is also preserved, as prescribed by Theorems 5 and 6.

### G.2    Most Benchmark BNs Have Few Transitive Edges

In this section we compute some attributes of 21 benchmark networks, which are publicly available at `http://compbio.cs.huji.ac.il/Repository/networks.html` and `http://www.bnlearn.com/bnrepository/`. These benchmark BNs contain the DAG structure and the conditional probability tables. Several prior works also used these BNs and evaluated DAG recovery by sampling data *observationally* by using the joint probability distribution [2, 30].

Table 2 reports the number of vertices, $|\mathrm{V}|$, the number of edges, $|\mathrm{E}|$, the number of transitive edges, $|\mathrm{RE}|$, and the ratio, $|\mathrm{RE}|/|\mathrm{E}|$. Finally, the mean and median of the ratios is presented. A median of $0.48\%$ indicates that more than half of these networks have a number of transitive edges less than $0.50\%$ of the total number of edges. In other words, our methods provide guarantees for exact learning of at least $99.5\%$ of the true structure for many of these benchmark networks.

### G.3    DAG Recovery on Benchmark BNs

In this section we test Algorithms 4, 5, 6, 7, 8, and 3, on benchmark networks that may contain transitive edges. The networks are publicly available at `http://www.bnlearn.com/bnrepository/`.

Figure 7: (Left, Top) Probability of correct structure recovery of the transitive reduction of a discrete CBN vs. number of samples per query, where the latter was set to $e^C \log nr$, with all CBNs having $r = 5$ and $\gamma \geq 0.01$. (Right, Top) Similarly, for continuous CBNs, the number of samples per query was set to $e^C \log n$, with all CBNs having $\|(\mathbf{I} - \mathbf{W})^{-1}\|_{2,\infty}^2 \leq 20$. (Left, Bottom) Results for imperfect interventions for discrete CBNs under same settings as in perfect interventions and $\alpha = 0.9$. (Right, Bottom) Results for imperfect interventions for continuous CBNs under same settings as in perfect interventions and $\nu_i^2 = \sigma_i^2, \forall i \in V$. Finally, we observe that there is a sharp phase transition from recovery failure to success in all cases, and the $\log n$ scaling holds in practice, as prescribed by Theorems 1, 2, 5, and 6.

Table 2: For each network we show the number of vertices, $|V|$, the number of edges, $|E|$, the number of transitive edges, $|RE|$, and the ratio, $|RE|/|E|$.

| Network | $|V|$ | $|E|$ | $|RE|$ | $|RE|/|E|$ |
|---|---|---|---|---|
| Alarm | 37 | 46 | 4 | 8.70% |
| Andes | 223 | 338 | 45 | 13.31% |
| Asia | 8 | 8 | 0 | 0.00% |
| Barley | 48 | 84 | 14 | 16.67% |
| Cancer | 5 | 4 | 0 | 0.00% |
| Carpo | 60 | 74 | 0 | 0.00% |
| Child | 20 | 25 | 1 | 4.00% |
| Diabetes | 413 | 602 | 48 | 7.97% |
| Earthquake | 5 | 4 | 0 | 0.00% |
| Hailfinder | 56 | 66 | 4 | 6.06% |
| Hepar2 | 70 | 123 | 16 | 13.01% |
| Insurance | 27 | 52 | 12 | 23.08% |
| Link | 724 | 1125 | 0 | 0.00% |
| Mildew | 35 | 46 | 6 | 13.04% |
| Munin1 | 186 | 273 | 1 | 0.37% |
| Munin2 | 1003 | 1244 | 6 | 0.48% |
| Munin3 | 1041 | 1306 | 6 | 0.46% |
| Munin4 | 1038 | 1388 | 6 | 0.43% |
| Pigs | 441 | 592 | 0 | 0.00% |
| Water | 32 | 66 | 0 | 0.00% |
| Win95pts | 76 | 112 | 8 | 7.14% |
| Average | | | | 5.46% |
| Median | | | | 0.48% |

These standard benchmark BNs contain the DAG structure and the conditional probability distributions. We sample data *interventionally* by using the manipulation theorem [21]. We then compare

the learned DAG versus the true DAG. Several prior works used these BNs and also evaluated DAG recovery by sampling data *observationally* by using the joint probability distribution [2, 30].

**Discrete networks.** We first present experiments on discrete BNs. For each network we set the number of samples $m = e^{12} \log nr$, and ran Algorithm 4 once. After learning the transitive reduction, we ran Algorithm 3 to learn the missing transitive edges. For the true edge set E and recovered edge set $\tilde{\text{E}}$, we define the edge precision as $|\tilde{\text{E}} \cap \text{E}|/|\tilde{\text{E}}|$, and the edge recall as $|\tilde{\text{E}} \cap \text{E}|/|\text{E}|$. The F1 score was computed from the previously defined precision and recall. As we can observe in Table 3, all of the networks achieved an edge precision of 1.0, which indicates that all the edges that our algorithm learned are indeed part of the true network. Finally, all networks also achieved an edge recall of 1.0, which indicates that all edges (including the transitive edges) were correctly recovered.

Table 3: Results on benchmark discrete networks. For each network, we show the number of nodes, $n$, the number of edges, |E|, the number of transitive edges, |RE|, the maximum domain size, $r$, the edge precision, $|\tilde{\text{E}} \cap \text{E}|/|\tilde{\text{E}}|$, the edge recall, $|\tilde{\text{E}} \cap \text{E}|/|\text{E}|$, and the F1 score.

| Network | $n$ | |E| | |RE| | $r$ | Edge precision | Edge recall | F1 score |
|---|---|---|---|---|---|---|---|
| Carpo | 60 | 74 | 0 | 4 | 1.00 | 1.00 | 1.00 |
| Child | 20 | 25 | 1 | 6 | 1.00 | 1.00 | 1.00 |
| Hailfinder | 56 | 66 | 4 | 11 | 1.00 | 1.00 | 1.00 |
| Win95pts | 76 | 112 | 8 | 2 | 1.00 | 1.00 | 1.00 |

**Additive Gaussian networks.** Next, we present experiments on continuous BNs. For each network we set the number of samples $m = e^C \log n$, and ran Algorithm 4 once. For the true edge set E and recovered edge set $\tilde{\text{E}}$, we define the edge precision as $|\tilde{\text{E}} \cap \text{E}|/|\tilde{\text{E}}|$, and the edge recall as $|\tilde{\text{E}} \cap \text{E}|/|\text{E}|$. The F1 score was computed from the previously defined precision and recall. As we can observe in Table 4, both networks achieved an edge precision of 1.0, which indicates that all the edges that our algorithm learned are indeed part of the true network. Finally, both networks also achieved an edge recall of 1.0, which indicates that all edges (including the transitive edges) were correctly recovered.

Table 4: Results on benchmark continuous networks. For each network, we show the number of nodes, $n$, the number of edges, |E|, the number of transitive edges, |RE|, the constant $C$, the maximum domain size, $r$, the edge precision, $|\tilde{\text{E}} \cap \text{E}|/|\tilde{\text{E}}|$, the edge recall, $|\tilde{\text{E}} \cap \text{E}|/|\text{E}|$, and the F1 score.

| Network | $n$ | |E| | |RE| | $C$ | Edge precision | Edge recall | F1 score |
|---|---|---|---|---|---|---|---|
| Magic-Irri | 64 | 102 | 25 | 11 | 1.00 | 1.00 | 1.00 |
| Magic-Niab | 44 | 66 | 12 | 7 | 1.00 | 1.00 | 1.00 |

### G.4 DAG Recovery on Real-World Gene Perturbation Datasets

In this section we show experimental results on real-world interventional data. We selected 14 yeast genes from the gene perturbation data in "Transcriptional regulatory code of a eukaryotic genome" [9]. A few observations from the learned BN shown in Figure 8 are: the gene YFL044C reaches 2 genes directly and has an indirect influence on all 11 remaining genes; finally, the genes YML081W and YNR063W are reached by almost all other genes.

Next we show experimental results on real-world gene perturbation data from Xiao et al. [33]. Figure 9 shows the learned DAGs for genes from mouses (Left) and humans (Right). For mouse genes we analyzed 17 genes and we can observe the following: the gene Spint1 reaches 3 genes directly and all other genes indirectly; finally, the genes Tgm2, Ifnb1, Tgfbr2 and Hmgn1 are the most influenced genes. For human genes we analyzed 17 genes and we observe the following: the gene CTGF reaches 1 gene directly and all the remaining genes indirectly; finally, the gene HNRNPA2B1 is reached by all genes.

Figure 8: DAG structure recovered from interventional data in [9]. The nodes correspond to yeast genes.

Figure 9: DAG structure recovered from interventional data in [33]. (Left) Nodes correspond to mouse genes. (Right) Nodes correspond to human genes.