[Reviews · NeurIPS 2018]

Reviewer 1



This paper studies the problem of learning causal Bayes nets from a specific type of single-node international queries. (Given this type of query access, it is information-theoretically impossible to learn the graph exactly -- but it is possible to learn its transitive reduction, as done in this paper.) The algorithm works in two steps: First, a noisy form of a path query is defined. A path query can answer whether there exists a path connecting two given nodes in the unknown graph. A simple algorithm is given to learn the graph using noisy path queries. Then it is explained how to simulate noisy path queries with single-node interventions. The authors also give some basic experimental evaluation of their algorithms and give sharper bounds for special cases of the problem. Overall, this seems like a reasonable paper that could be accepted to NIPS. One issue I had with the paper is that the parameters are a bit confusing and it is not immediately clear how tight the results are. The authors give some related references, but it would be useful to elaborate somewhat on the precise statements -- if only in the appendix.

Reviewer 2



This paper gives algorithms for recovering the structure of causal Bayesian networks. The main focus is on using path queries, that is asking whether a direct path exists between two nodes. Unlike with descendant queries, with path queries one could only hope to recover the transitive structure (an equivalence class of graphs). The main contribution here is to show that at least this can be done in polynomial time, while each query relies on interventions that require only a logarithmic number of samples. The author do this for discrete and sub-Gaussian random variables, show how the result can be patched up to recover the actual graph, and suggest specializations (rooted trees) and extensions (imperfect interventions). The paper is generally well written. While the main result is not terribly deep or surprising (that transitive structure is easier to recover and that logarithmic number of samples are enough for an acceptable query quality), the general ideas and treatment are novel to me and relevant to the community. I have not found any major issues with the portions of the theory that I looked at (though, as a disclaimer, I did not go over every bit of it). More importantly, the proposed algorithms are intuitive and simple enough to be useful, the analysis is mostly straightforward, and this line of inquiry also opens up many new questions. I recommend the paper for acceptance. Here are some specific major remarks. - Being able to do something simpler when the more complicated thing is computationally hard is generally interesting. But I would be much more motivated if the simpler solution is in itself useful. Are there situations where the transitive structure in itself is useful? - We do not quite see the topology of the graph explicitly influencing anything except when we talk about rooted trees. This is somewhat strange. And indeed there are many implicit spots where the topology plays a role and which I think are important to highlight. The most important of these is when the authors claim (L265-266) that "the sample complexity for queries [...] remains polynomial in n" when answering transitive queries. Now think of a sequence of graph families, indexed by n, where the mean number of parents increases linearly, while the influence of any given parent gets diluted the more parents a node has. Then, for a sequence within such families, \gamma as defined in Theorem 3 (L268) would decay, possibly even exponentially since there are exponentially many S in the number of parents, therefore invalidating the claim. Conversely, if one assumes that \gamma stays constant in the asymptotics, then one in effect is making the claim only for a specific sequence of graph families. I think this kind of implicit topological assumptions should be much more transparent in the final write-up of the paper. Here are more minor remarks. - Put the word "causal" (Bayes nets) in the title, it is misleading otherwise. - The first time n is used (L52), it's not yet defined (unless I'm missing it). - The \tilde O notation just for constants is strange (footnote 2). Why not just O? - L111, Why not just say p(Y) in the |Dom[Y]| simplex? That would be more accurate. Perhaps write \Delta_{Dom[Y]}. Also if you go through that much trouble to describe discrete random variables, just saying one line for continuous ones is strange, as these can be much trickier. At least say that you assume some densities with respect to a common measure. - L138, the only time you do not require correctness is when there is a directed path between i and j *and* i is not a parent of j. - L170, the remark "Under this viewpoint..." is a bit confusing. It makes allusion to using a topology that is not a priori known. Or perhaps are you saying if the topology is known to be in a restricted family, then one could hope for reduced effort? Either way, it's worth clarifying. - Generally, it would be nice for propositions/theorems to invoke the subset of the assumptions that they use. - L185, it should be clear that using the expectation can generally miss out on the probability discrepancies, unless the expectation (or expectations) are sufficient statistics. - L189, I think it's more accurate to say that the statistics are inspired from Propositions 1&2, while the needed threshold is inspired from Theorems 1&2. - A lot hinges on knowing lower bounds on the threshold / upper bounds on variances. Is there any work on adapting to them being unknown? (Beyond the remark on L203.) - Please clarify the role of S in Definition 5. The text on L252-254 is not adequate, especially that the proofs of Theorems 3 and 4 hand-wave too much and refer back to those of Theorems 1 and 2. As I see it, either S shouldn't even be part of the definition or we should say that the \delta-noisiness property should hold for some choice of S. - In Algorithm 3, just say the input is the output of Algorithm 1 (to be consistent to the main text). - L297, at most \epsilon. - Would Definition 6 help in the general setting? - L309, it's inaccurate to say the sample complexity for sub-Gaussian random variables under imperfect interventions is unchanged. As I understand it, the effective variance changes. Please clarify. [After author feedback] I thank the authors for their detailed answers. My assessment is unchanged, though I raised my confidence.

Reviewer 3



Authors use single vertex interventions and declare the variables whose distribution changes with the intervention to be the descendants. Based on this, they can recover the transitive reduction of a causal graph with n interventions. The advantage of this approach, although it increases the number of interventions from log^2(n) to n compared to [13], is that it requires much less number of samples, since intervention sets are all of size 1. The fact that an intervention changes the distribution of the descendants even in the presence of latents has been known and is lately used in [13]. Accordingly, the overall approach is not surprising, when the goal is to minimize the number of interventional samples. This approach is presented using what the authors call "a path query". What the authors call a path query seems equivalent to a conditional independence test on the post-interventional distribution when we use a stochastic intervention. Algorithm 3, the algorithm for recovering the true causal edges, gradually increments the parent set of each node by including the newfound parent to the current parent set. This algorithm however requires exponential number of samples, and its advantage is not clear to me over the algorithms of [13]. Given this high level idea, authors analyze the number of samples required to reliably run these tests. I am happy to see this analysis and this is definitely an important direction for the causality literature. However the presented analysis have some issues. Most importantly, the authors use DKW inequality, which is useful for continous variables, in the proof of Theorem 1. However, since they are using empirical counts to approximate a discrete distribution, they could have used Chernoff bounds. In fact, had they used Chernoff bounds, I believe they would get a 1/sqrt(gamma) dependence, whereas with the current analysis, they have a 1/gamma^2 dependence. (details below) Another important issue with the submission is the title: The title is very misleading. The authors only find the transitive reduction of a causal graph. Not learn the whole Bayesian network. I recommend mentioning causal graph and transitive reduction in the title. It is also surprising to see that the authors did not choose causality as primary or secondary topic for the submission. Despite the not surprising algorithm and some issues with the analysis, I am leaning more towards acceptance. However, I would appreciate if the authors could respond to the questions I asked in the following and clarify certain points for me to make a better assessment. Following are more details and my feedback/other questions for the authors: What authors call a path query is simply checking if a descendants distribution changes or not. I understand that they are following up on [32], but calling this known test a "path query" makes the work seem disconnected from the existing causality literature. I would remove this language from the paper since it is simply checking if the underlying distribution of the other nodes changes with different interventions. Also, the test seems equivalent to a CI test, when we use a stochastic intervention on the support of the intervened node. Could you please comment on this and maybe add a discussion about this equivalence in the paper? There are certain issues with the way background is explained. The authors state that Bayesian networks are also used to describe causal relations. hen they are called causal Bayesian networks. This is not correct. A Bayesian network is a graph according to which the joint distribution factorizes. Bayesian networks do not have causal interpretations. Please change the narrative accordingly in line 24 and also in line 100, where it is told that a Bayesian network can also be viewedd as a causal model or a causal Bayesian network. This narrative is misleading. The definition of a v-structure in footnote 1 in page 1 should mention the fact that X, Z should not be adjacent. The authors use "number of interventions" for the number of total experiments performed in order to realize a collection of interventions: If they perform a single intervention on a 5 state variable, they count this as 5 interventions. This arises slight confusion, especially when they are citing the performance of the related work. I would use "number of interventional samples" for this to avoid confusion. But this is not a major point. In line 108, the probability expression should contain Xi as well, since authors use indicator Xi=xi on the right hand side. The path query definition given in lines 120-122 is word by word taken from [32], which the authors cite elsewhere in the paper. Please add this citation here in the definition to signify where the definition is taken from. In line 149, authors mention that their method is robust to latent confounders. This is correct since they only use interventional data and search for the desendants by looking at the invariances in the interventional distribution (This is in fact how I would describe this work and the algorithm rather than using the path query language. This would also make the paper more consistent with the existing causality literature). However, it is too important to simply mention in a single line. Please elaborate and explain why latent confounders do not affect any of the analysis in the paper (no need to respond here). Proof of proposition 2 is not rigorous. Authors should use faithfulness. Please see the appendix of [13] for a similar proof. (I suspect interventional faithfulness to be necessary here). Please comment on this. In Theorem 1, the authors simply estimate each interventional distribution from the empirical counts, and check if they are different enough. The analysis of this algorithm uses what is known as DKW inequality. This inequality is useful for continuous random variables. I don't understand why the authors have not used a Chernoff boumd. I believe with Chernoff bound, they should be able to get a 1/sqrt(gamma) bound, whereas with this inequality, they can only show 1/gamma^2 bound. The only change in the analysis would be that, since the difference in distributions is assumed to be gamma, you need a concentration of gamma/2 in each distribution. But this would only change the constant. Please check the analysis to verify this suggestion. In Theorem 2, the authors assume that Xj are subGaussian. But how do we know P(Xj|do(Xi)) are also subGaussian? Please explain. Because the proof of theorem 2 uses this. The authors should state this as an assumption of Theorem 2. This seems necessary for this result. Algorithm 3 operates by, for each node adding the parental nodes not already included in the transitive reduction. The algorithm seems correct. However, with the authors' definition of the number of interventions, this requires exponential in n number of interventions since they have to intervene on all the parents of every node. They acknowledge this elsewhere in the paper, but not including this fact in the theorem statement is a slighly misleading. It is not clear what the advantage of Algorithm 3 is, compared to the algorithms given in [13]. Please elaborate. There are various grammatical errors and typos throughout. I recommend proofreading the submission. Following are what I could catch: Line 5: any causal Bayesian networks -> any causal Bayesian network Line 8: by intervening the origin node -> by intervening on the origin node Line 23: are a powerful representation -> are powerful representations Line 33: the effect on Y by changing X -> the effect of changing X on Y Line 34: DAGs -> causal graphs Line 40: or assume that have one -> or assume that they have one Algorithm 2, Line 1: Intervene Xi -> Intervene on Xi